# Instrument Artifacts Lead to Uncertainties in Parameterizations of Cloud Condensation Nucleation

**Jessica A. Mirrielees and Sarah D. Brooks**

Texas A&M University, College Station, TX 77843

Correspondence E-mail: jmirrielees@tamu.edu

**Abstract**
The concentrations of cloud condensation nuclei (CCN) modulate cloud properties, rainfall
location and intensity, and climate forcings.  This work assesses uncertainties in CCN
measurements and the apparent hygroscopicity parameter ($\kappa_{app}$) which is widely used to represent
CCN populations in climate models. CCN measurements require accurate operation of three
instruments: the CCN instrument, the differential mobility analyzer (DMA), and the condensation
particle counter (CPC).  Assessment of DMA operation showed that varying the ratio of aerosol to
sheath flow from 0.05 to 0.30 resulted in discrepancies between the  $\kappa_{app}$ values calculated from
CCN measurements and the literature value. Discrepancies were found to increase from < 1% to
13% for both sodium chloride and ammonium sulfate.  The ratio of excess to sheath flow was also
varied, which shifted the downstream aerosol distribution towards smaller particle diameters (for
excess flow < sheath flow) or larger particle diameters (for excess flow > sheath flow) than
predicted.  For the CPC instrument, undercounting occurred at high concentrations, resulting in
calculated $\kappa_{app}$ lower than the literature values.  Lastly, undercounting by CCN instruments at
high concentration was also assessed, taking the effect of supersaturation on counting efficiency
into account.  Under recommended operating conditions, the combined DMA, CPC, and CCN
uncertainties in $\kappa_{app}$ are 1.2 %  or less for 25 to 200 nm diameter aerosols.
**Copyright Statement**
Will be provided by Copernicus.

## 1. Introduction

Aerosol-cloud interactions represent a major uncertainty in current predictions of the Earth's climate (IPCC, 2013). According to well-known Köhler theory, an aerosol's potential to catalyze cloud droplet formation by activating as a cloud condensation nucleus (CCN) depends on its physical and chemical properties. For any given composition, the CCN activation potential of an aerosol increases as its diameter decreases. While the relationship between aerosol diameter and CCN activation is straightforward, the effect of composition on an aerosol's ability to participate in cloud formation is more complex (Petters and Kreidenweis, 2013; Ovadnevaite et al., 2011). Predicting the cloud forming capacity of various air masses based on the properties of the aerosol they contain is essential for evaluating relative contributions from pollution, continental background and marine aerosol sources (Brooks and Thornton, 2018; Carslaw et al., 2013). Long-term CCN measurements are available from numerous locations globally (Schmale et al., 2018). However, understanding regional and temporal variability in CCN populations requires the ability to assess whether observed differences reflect true physical differences or simply variations in CCN sampling strategies.

Parameterizations of CCN activity which accurately prescribe CCN measurements are needed for climate models, cloud resolving models, and air quality predictions (Betancourt and Nenes, 2014; Betancourt et al., 2013; Chang et al., 2017; Crosbie et al., 2015; Karydis et al., 2012; Kawecki and Steiner, 2018). One parameterization was designed to represent the cloud droplet activation potential ambient aerosol masses of unknown composition with a single variable, kappa ($\kappa$) based on the dry aerosol's hygroscopicity, or ability to uptake water and form a solution droplet (Petters and Kreidenweis, 2007). Various names and abbreviations have been given to $\kappa$ throughout the literature: "hygroscopicity parameter", "single hygroscopicity parameter", $\kappa$ (Petters and Kreidenweis, 2007;

Carrico et al., 2008; Asa-Awuku et al., 2010; Moore et al., 2012b); "CCN-derived $\kappa$", $\kappa_{CCN}$ (Carrico et
al., 2008; Petters and Kreidenweis, 2007); and the "apparent hygroscopicity parameter" $\kappa_{app}$ (Sullivan
et al., 2009; Collins et al., 2016; Petters and Kreidenweis, 2013). The term *apparent* hygroscopicity is
favored by many because it emphasizes that fact that while CCN activation can often be predicted
accurately by hygroscopic water uptake, they are different physical processes. It is possible for a
compound to have high intrinsic hygroscopicity and low apparent hygroscopicity if it is poorly soluble
in water (Sullivan et al., 2009).
Parameterizations of hygroscopicity that pre-date Petters and Kreidenweiss 2007 exist as well. Winkler
1973 developed an equation for approximating the growth of an aerosol particle with relative humidity,
based on the quantity and physical characteristics of the soluble species in the particle. Another
approximation for the relationship between the equilibrium size of a particle and relative humidity was
derived by Fitzgerald in 1975, in which the soluble fraction and composition of the soluble
component(s) are taken into account. Fitzgerald et al., 1982 derived a particle composition parameter
using the mass fraction and physical properties of soluble material in a particle. Kreidenweis et al., 2005
determined that the critical activation diameter of dry aerosol particles can be calculated from simplified
Köhler theory using the physical properties of water and the solute in a solution droplet. This
parameterization has been used in CCN closure studies (Bougiatioti et al., 2009; Moore et al., 2011;
Moore et al., 2012a). The earliest prediction of CCN concentrations for specific particle diameters and
hygroscopicity used this parameterization as well (Mochida et al., 2006).

Once calculated, hygroscopicity parameters are useful tools for comparing CCN field measurements
conducted in various regions and seasons and for making predictions about cloud formation, aerosol-
cloud interactions in weather, and climate models. Values of $\kappa_{app}$ can be used to compare the CCN
results in field and laboratory studies, including sea spray aerosol.  For example, aggregation of results
from several mesocosm experiments and marine field studies found submicron (30-100 nm) $\kappa_{app}$ for sea
spray aerosol as low as 0.4 and as high 1.3 (Collins et al., 2016).  Another study, which included a
survey of observational CCN data, reported that marine and continental aerosols could be described by
$\kappa_{app}$ values of $0.7 \pm 0.2$ and $0.3 \pm 0.1$ respectively (Andreae and Rosenfeld, 2008).

Several studies have examined the sensitivity of models to $\kappa$ values derived from HTDMA
measurements. An analysis of the NASA Global Modeling Initiative Chemical Transport Model and the
GEOS-Chem CTM (Karydis et al., 2012) found that cloud droplet number concentration is sensitive to $\kappa$
in Arctic and remote regions, where background aerosol loadings are low.  Another study (Betancourt
and Nenes, 2014) found that a $\pm50$ % uncertainty range in the $\kappa$ of secondary organic aerosols and
particulate organic matter resulted in a cloud droplet number concentration uncertainty of up to 15 %
and 16 %, respectively.  Updating precipitation models with lab-derived $\kappa$ values for specific inorganic
and organic species may increase the accuracy of storm forecasts by providing better predictions of
intense precipitation (Kawecki and Steiner, 2018).  In terms of climate, (Liu and Wang, 2010) found that
increasing the $\kappa$ of primary organic aerosols from 0 to 0.1, and decreasing the $\kappa$ of secondary organics
aerosols from 0.14 to 0.07, resulted in an uncertainty in global secondary aerosol indirect forcing of 0.4
Wm$^{-2}$ from pre-industrial times to present day.

The sensitivity of weather and climate models to hygroscopicity parameters demonstrates the need for
accurate measurements. In this study, we examine experimental uncertainties in CCN measurements and
the resulting uncertainties in determination of $\kappa_{app}$.  Differences in reported $\kappa_{app}$ values may result
from experimental artifacts rather any actual differences in an aerosol's ability to facilitate cloud
formation.  By systematically quantifying sources of experimental error, this study provides a
framework for determining the significance of variations in CCN properties reported in multiple studies
and defining the operating conditions which minimize instrumental artifacts.
**2. Background**

The Köhler equation relates water vapor saturation ratio at the surface of a wet droplet, $s$, to its radius at
equilibrium (Rogers and Yau, 1989):

$$s = \left(1 - \frac{b}{r^3}\right) \exp\left(\frac{a}{r}\right) \tag{1a}$$


$$a = \frac{2\sigma_w M_w}{\rho_w RT} \tag{1b}$$


$$b = \frac{3 i m_s M_w}{4\pi \rho_w M_s} \tag{1c}$$


where $s$ is the equilibrium saturation ratio of a solution droplet with radius $r$, $\sigma_w$ is the surface tension of
water, $M_w$ is the molecular weight of water, $R$ is the ideal gas constant, $T$ is temperature in Kelvin, $\rho_w$ is
the density of water, and $M_s$ is the molecular weight of the solute. The minimum saturation ratio that is
required for spontaneous droplet growth, $s_{act}$, is therefore:

$$s_{crit} = 1 + \sqrt{\frac{4a^3}{27b}} \tag{2}$$


Petters and Kreidenweis [2007] reformulated the Köhler equation as κ-Köhler theory:

$$s_{crit} = exp\left(\sqrt{\frac{4A^3}{27 D_{act}^3 \kappa_{app}}}\right) \tag{3a}$$

and

$$A = \frac{4\sigma_{lv}M_w}{RT\rho_w} \quad (3b)$$


Where $s_{crit}$ is the critical water vapor saturation ratio, $D_{act}$ is the dry particle activation diameter and
$\kappa_{app}$ is the apparent hygroscopicity parameter. Solving for $\kappa_{app}$ yields:

$$\kappa_{app} = \frac{4A^3\sigma_{lv}^3}{27T^3D_{act}^3\ln^2(s_{crit})} \quad (4)$$


The apparent hygroscopicity parameter can be calculated from experimental CCN results, where the dry
diameter and water vapor saturation ratio are known. For a chosen aerosol diameter, the activated
fraction is the ratio of the concentration aerosols that activate as CCN to the total aerosol concentration:

$$Activated\ fraction = \frac{CCN\ Concentration}{Aerosol\ Concentration} \quad (5)$$


Activated fraction data is fit with a sigmoid error function to determine the percent supersaturation at
which 50 % of the particles have activated as CCN (activated fraction = 0.50), which is considered the
operationally defined critical percent supersaturation $SS_{crit}$ (Rose et al., 2008). The critical saturation
ratio $s_{crit}$ can then be determined and entered into Eq. (4) in order to calculate $\kappa_{app}$ for the near-
monodisperse aerosol:

$$s_{crit} = 1 + \frac{SS_{crit}}{100} \quad (6)$$


Reporting $\kappa_{app}$ as a function of diameter allows for the comparison of the cloud condensation nucleation
abilities of multimodal aerosol populations, without overlooking differences which arise due to aerosol
composition.

The apparent hygroscopicity parameter is related to chemical composition; therefore, the calculated $\kappa_{app}$
of a pure substance should be constant across CCN experiments.  However, discrepancies between $\kappa_{app}$
for a single chemical species have been observed. Experimental results for ammonium nitrate are
inconsistent with reported values ranging from $0.577 \leq \kappa_{app} \leq 0.753$ (Svenningsson et al., 2006).
Also, large ranges are often observed for organic compounds, such as glutaric acid ($0.054 \leq \kappa_{app} \leq$
$0.16$) and malonic acid ($0.199 \leq \kappa_{app} \leq 0.255$) (Koehler et al., 2006; Kumar et al., 2003; Hartz et al.,
2006).  Below we evaluate potential sources of uncertainties in CCN measurements and the resulting
uncertainties in $\kappa_{app}$.

**3. Artifacts derived from sized CCN measurements**

CCN measurements used for calculating apparent hygroscopicity from monodisperse aerosol require accurate operation of three instruments: the CCN, the differential mobility analyzer (DMA), and the condensational particle counter (CPC). The setup for laboratory CCN experiments is shown in Fig. 1. First, a polydisperse population of aerosols is generated by an atomizer and dried using a desiccant tube packed with silica gel. A near-monodisperse flow is obtained through size-selection in the DMA. The flow is then split between a CPC (which measures aerosol concentration) and a CCN counter (which measures the concentration of particles that activate as cloud condensation nuclei at a given percent supersaturation). Instrument artifacts will first be assessed separately for the DMA, CPC, and CCN counter. In the concluding section of the paper (and Fig. 10), the overall uncertainty due to the combination of these is presented and discussed.

We note that this study considers sized CCN measurements which may be used for the determination of $\kappa_{app}$. In contrast, a number of earlier CCN studies conducted on the full ambient aerosol population without sizing the aerosol (Jennings et al., 1996; Hudson and Xie, 1998; Modini et al., 2015; Duan et al., 2017; Schmale et al., 2018; Leng et al., 2013). While useful, such studies do not produce the data required for accurate determination of $\kappa_{app}$ from the CCN measurements.

**3.1 Artifacts derived from differential mobility analyzers**

**3.1.1   DMA operation and electrical mobility**

Differential mobility analyzers used in atmospheric science include commercially available instruments
from Grimm Aerosol Technik, TSI Incorporated, and MSP Corporation. They have also been custom
built by a number of research groups (Mei et al., 2011;Barmpounis et al., 2016;Jokinen and Makela,
1997;Seol et al., 2000). All models allow for the selection of particles through electrical mobility, the
ability of a particle to move through a medium (such as air) while acted upon by an electrical field. The
DMA size-selects near-monodisperse aerosol from a polydisperse aerosol source, as shown in Fig. 2
(modeled after the Vienna-type long Differential Mobility Analyzer from Grimm Technologies). The
electrical mobility $Z_p$ of a particle with mobility diameter $d_m$ can be calculated according to:

$$Z_p = \frac{neC_C(d_m)}{3\pi\eta d_m} \tag{7}$$


where $n$ is the number of charges on the particle (assumed to be one in this study), $e$ is the elementary
unit of charge, $\eta$ is the gas dynamic viscosity, and $C_C(d_m)$ is the Cunningham slip correction factor:

$$C_C(d_m) = 1 + \frac{2\lambda}{d_m}\left(\alpha_{CC} + \beta_{CC}\exp\left[-\frac{\gamma_{CC}}{2\lambda/d_m}\right]\right) \tag{8}$$


where $\lambda$ is the mean free path (DeCarlo et al., 2004). For the Vienna-type long Differential Mobility
Analyzer from Grimm Technologies, Inc. considered here, $\alpha_{CC} = 1.246$, $\beta_{CC} = 0.42$, and $\gamma_{CC} = 0.86$
(Grimm Aerosol Technik, 2009).

Particle-laden flow enters the differential mobility analyzer through the aerosol inlet (flow $Q_a$), and travels
down the DMA column (inner radius $r_1$, outer radius $r_2$) with the clean air sheath flow $Q_{sh}$. Positively-
charged particles are attracted by the negatively-charged inner electrode, to which voltage $V_0$ has been
applied. Ideally, selection of a voltage allows only particles of a specific mobility diameter to exit the
DMA through the sample flow $Q_s$. All particles with larger diameter (lower $Z_p$) or smaller diameter
(higher $Z_p$) will exit the DMA through the excess flow $Q_e$. In other words, $Q_s$ would ideally consist only
of aerosols with diameters equal to, or very nearly equal to, the selected diameter.

In reality, the aerosol flow that leaves the DMA through $Q_s$ is polydisperse with a mobility distribution
determined by instrumental parameters. A triangular approximation has been chosen as a model for this
distribution, as particle inertia is negligible for the diameters considered in this study (Stratmann et al.,
1997; Mamakos et al., 2007). The probability that a particle at the aerosol inlet will exit with the sampling
flow is defined by transfer function $f\left(Z_p, Z_{p,mid}\right)$:

$$f\left(Z_p, Z_{p,mid}\right) = \frac{\alpha_{TF}}{2\beta_{TF}}\left(\left|\frac{Z_p}{Z_{p,mid}} - (1 + \beta_{TF})\right| + \left|\frac{Z_p}{Z_{p,mid}} - (1 - \beta_{TF})\right| - 2\left|\frac{Z_p}{Z_{p,mid}} - 1\right|\right) \quad (9)$$

where $Z_{p,mid}$ is the midpoint mobility of the transfer function, and $\alpha_{TF}$ and $\beta_{TF}$ are flow-derived
constants, defined as:

$$\alpha_{TF} = \frac{Q_s + Q_a}{2Q_a} \quad (10a)$$

and
$$\beta_{TF} = \frac{Q_s}{Q_{sh}} \quad (10b)$$


The midpoint and half-width of the transfer function are respectively calculated according to: (Knutson
and Whitby, 1975)
$$Z_{p,mid} = \frac{Q_e + Q_{sh}}{4\pi L V_0}\ln\left(\frac{r_2}{r_1}\right) \quad (11a)$$

and

$$\Delta Z_p = \frac{Q_a}{2\pi L V_0} \ln\left(\frac{r_2}{r_1}\right) \tag{11b}$$


where L is the distance between the DMA inlet and outlet.

**3.1.2 $\kappa_{app}$ artifacts arising from DMA flow ratios**

Next we assess the ramifications of the DMA transfer function for the derived $\kappa_{app}$. A lognormal
theoretical aerosol number distribution was used to represent a polydisperse ambient aerosol population
(Fig. 3a). This distribution was converted to an electrical mobility distribution using Eq. (7) and Eq. (8),
assuming that the aerosols in the distribution were spherical and singly charged. From the distribution, a
series of single aerosol sizes were selected (25, 50, 100, and 200 nm diameter). For each aerosol size,
the resulting DMA transfer functions were calculated for seven cases using Eq. (9) and the various
parameters for DMA sheath, excess, aerosol, and sample flow listed in Table 1. These seven cases were
chosen to represent possible measurements scenarios that may be encountered in a CCN experiment.
The aerosol/sheath ratio is varied in Cases 1-4 in order to study the effects of chosen experimental
parameters. Sheath flow is predetermined in some DMAs (for example, the Grimm Vienna DMA
considered in this study), but can be varied in other instruments. The aerosol flow rate may also be
selected in an experiment. Cases 5-7 vary the excess/sheath ratio in order to take proper instrument
operation into account. The excess and sheath flow should be identical, but small discrepancies may
occur.

For example, the resulting DMA transfer functions for a 100 nm aerosol conditions constrained by Cases
1-4 are shown in Fig. 3b, where an increase in $Q_a/Q_{sh}$ from 0.1 (black line) to 0.3 (green line) tripled
the width of the number distribution, and decreasing $Q_a/Q_{sh}$ to 0.05 (blue line) from 0.10 halved the
width of the number distribution. The result of applying the transfer functions shown in Fig. 3b to the
distribution in Fig. 3a is shown in Fig. 3c.

All downstream distributions for all seven DMA cases and all aerosol sizes are shown in Fig. S1 in the
Supplement.  DMA Cases 1-4 represent experimental conditions in which the sheath and excess air
flows are equal and the aerosol/sheath flow ratio is varied.  As $Q_a/Q_{sh}$ increases, the width of the
number distribution measured downstream of the DMA increases, while the midpoint diameter remains
constant.  It was found that doubling the aerosol to sheath ratio doubled the width of the downstream
number distribution for 25, 50, 100, and 200 nm particles.  For example, when selecting 200 nm
particles, increasing $Q_a/Q_{sh}$ from 0.10 to 0.20 increased the downstream diameter range from 181-222
nm (a spread of 41 nm) to 167-250 nm (a spread of 87 nm).  The particle diameter ranges that would be
observed downstream of the DMA are summarized in Table 2.

To assess the variations in CCN properties resulting from DMA uncertainties the critical percent
supersaturation were calculated for representative atmospheric aerosols. The value of  $SS_{crit}$ was
calculated for each particle diameter using Eq. (3a), using literature values for  apparent hygroscopicity
of 0.61 for ammonium sulfate and 1.28 for sodium chloride (Clegg et al., 1998).  It should be noted that
this analysis considers two homogeneous aerosol distributions of hygroscopic salts.  Real aerosol
distributions tend to be mixtures of many species, and the shape of the number distribution can vary
between species.

To test how uncertainties in DMA diameter translate to uncertainties in $\kappa_{app}$, the activation of particles
downstream of the DMA was assessed. First, for each case and diameter (25, 50, 100, and 200 nm) the
critical saturation ratio $s_{crit}$ was calculated for each particle diameter range downstream from the DMA
using Eq. 3a. These critical saturation ratios were converted to critical percent supersaturation $SS_{crit}$
and used to calculate the activated fraction $AF$ for the aerosol particles downstream from the DMA for
percent supersaturations $0.01 < SS < 1.5$, using the equation:

$$AF = \frac{1}{2}\left(1 + \text{erf}\left(\frac{SS - SS_{crit}}{\sigma\sqrt{2}}\right)\right) \qquad (12)$$

where the standard deviation $\sigma$ was equal to one-hundredth of $SS_{crit}$. The small $\sigma/SS_{crit}$ ratio was
chosen in order to generate accurate activated fraction curves for each particle diameter.

The activated fraction curve for each selected diameter (25, 50, 100, and 200 nm) was then calculated as
the sum of the number-weighted activated fractions of each particle diameter downstream from the
DMA. For example, for a selected diameter of 25 nm, the downstream diameters ranged from 23 nm to
27 nm for DMA Case 1 and from 20 nm to 36 nm in DMA Case 4. The equation used for this
calculation is:

$$AF_{weighted} = \sum_i \frac{n_i}{n_{total}} AF_i \qquad (13)$$

where $AF_i$ is the activated fraction calculated using Eq. 12 and $\frac{n_i}{n_{total}}$ is the fraction of particle
downstream from the DMA of diameter $i$.

This calculation was repeated for each selected diameter (25, 50, 100, and 200 nm), each DMA Case (1-
7), and percent supersaturation (0.01-1.5) in order to construct activation curves for each selected
diameter and DMA Case.   As an example, in Fig. S2, the shape and position of each activated fraction
curve vary with the DMA flow ratios.  As the aerosol/sheath ratio increases, the activated fraction curve
flattens out (DMA Case 4). The critical percent supersaturation $SS_{crit}$ was then determined for each
activation curve as the percent supersaturation where $AF = 0.50$.  These results are shown in Fig. 4a
for ammonium sulfate and sodium chloride.  Eq. 4 was then used to calculate $\kappa_{app,theory}$ for each DMA
Case and selected diameter, as shown in Fig. 4b.  Discrepancies between $\kappa_{app,theory}$ calculated in this
study and literature values (hereon referred to as "$\kappa_{app}$ artifacts") are shown for both compounds in Fig.
4c-d.

The largest $\kappa_{app}$ artifact was found in DMA case 4 (where the aerosol/sheath ratio was the highest) for
both ammonium sulfate and sodium chloride aerosols.  The artifacts for 25 nm ammonium sulfate
aerosol in DMA case 4 was 0.08 , or ~13% of the literature value used for $\kappa_{app}^{(NH_4)_2SO_4}$, while the artifacts
for 25 nm sodium chloride in DMA case 4 was 0.16, or ~13% of the literature value used for $\kappa_{app}^{NaCl}$.
Artifacts were also high for DMA case 6 ($-0.041 \leq \kappa_{app,artifact}^{(NH_4)_2SO_4} \leq -0.048$) and DMA case 7
($0.014 \leq \kappa_{app,artifact}^{(NH_4)_2SO_4} \leq 0.024$), where sheath and excess flow were unequal.  This result demonstrates
that artifacts may still occur when low aerosol/sheath flow ratios are chosen (0.15 and 0.08 for DMA
cases 6 and 7, respectively) due to small differences between sheath and excess flow rates (5% and 2%
for DMA cases 6 and 7, respectively).

$\kappa_{app}$ artifacts were larger for sodium chloride ($-0.10 \leq \kappa_{app,artifact}^{NaCl} \leq 0.16$) than for ammonium
sulfate ($-0.05 \leq \kappa_{app,artifact}^{(NH_4)_2SO_4} \leq 0.08$) across the DMA cases.  As our results show, when two or more
compounds are compared, the more hygroscopic compound will have larger $\kappa_{app}$ artifacts.

This analysis was also applied to the range of apparent hygroscopicity values Svenningsson et al., 2006
reported for ammonium nitrate  $0.577 \leq \kappa_{app} \leq 0.753$, with a mean value of 0.670.  If 0.670 is
assumed to be the true $\kappa_{app}$ for ammonium nitrate, then the sample/sheath ratio used to determine $\kappa_{app}$
(1.2-2.0 L min$^{-1}$) could lead an experimental kappa as low as 0.665 or as high as 0.674, which would not
fully explain the actual experimental range. This assessment ignores possibility of under/over counting
which could introduce additional errors.

In addition to the errors discussed above, accuracy in CCN measurements depend on the accuracy of the
instrument calibration.  Specifically, accurate determination of the percent supersaturation set points
within the CCN instrument are dependent on accurate sizing of aerosols entering the CCN, and therefore
are dependent on the DMA sizing during CCN calibration. CCN calibrations during two standard
compounds, ammonium sulfate and sodium chloride, as described in detail in Rose (2008). Fortunately,
if the calibration procedure described by Rose is followed and an optimal DMA aerosol to sheath ratios
employed, the uncertainties will be minimal. Specifically, this analysis shows that an aerosol/sheath
ratio of 1:10 or 1:20 (Case 1 or 2, respectively) is recommended for all CCN calibrations.  This will
result in $\kappa_{app}$ uncertainties of less than 1% for all dry sizes (25 to 200 nm).  However, if CCN
calibrations are performed using a DMA operated with less than ideal aerosol to sheath ratios,
substantial errors will be introduced.  Analysis of the impact of DMA uncertainties on CCN calibrations
are discussed in detail in the Supplemental Materials. In the worst case scenario amongst the cases
evaluated here (Case 4), the resulting uncertainty in $\kappa_{app}$ is 15%.

**3.1.3   Effect of double and triple charges on particles**

During normal operation, the Grimm DMA employs a bipolar charger (also known as a neutralizer) to
charge aerosol particles through the capture of gaseous ions.  The analysis in Section 3.1.2 assumes that
each particle carries a single (+1) charge.  In reality, the methods used to charge particles prior to
entering a DMA may impart two, three, or more charges to individual particles (Fuchs, 1963).  The
charge distribution resulting from a bipolar charger is roughly approximated using the Boltzmann law
(Keefe et al., 1959).   However, the Boltzmann law assumes symmetric aerosol particle charging (equal
concentrations of negatively and positively charged particles).  Deviation from symmetric charging is
observed in regions of high ionizations, and this deviation becomes more pronounced as particle size
increases (Hoppel and Frick, 1990).  A more accurate estimation of stationary charge distribution has
been calculated using an approximation formula for the charge distribution produced by a bipolar
charger:

$$f(k) = 10^{\left[\sum_{i=0}^{i=5} a_i(k)(\log_{10} D_{nm})^i\right]}$$
(14)


where $f(k)$ is the fraction of particles carrying $k$ charges, $a_i(k)$ are approximation coefficients
determined using a least-squares regression analysis, and $D_{nm}$ is the particle diameter in nanometers
(Wiedensohler, 1988).  The approximation coefficients only apply to particles with 0, ±1, and ±2
charges.  In a separate study, Maricq et al., 2008 determined approximation coefficients for poly ($\alpha$-
olefin) oligomer oil droplets with ±1, ±2, and ±3 charges.  The approximation coefficients reported by
these two studies were in excellent agreement for particles with ±1 and in weak agreement for ±2
charges (+2 and -2 charging efficiencies were overestimated by 50% and 100%, respectively).
Therefore, this analysis will use the approximation coefficients from Wiedensolher, 1988 for particles
with +1 and +2 charges, and the approximation coefficient for particles with +3 charge from Maricq et
al., 2008.

In order to assess the impact of multiple charges on $\kappa_{app}$, Eq. (14) and the approximation coefficients
from Wiedensolher, 1988 and Maricq et al., 2008 were used to calculate the charge distribution of the
representative aerosol population shown in Fig. 3a.  The resulting charge distribution is shown in Fig.
S6a.  An increase in multiple charging is observed as particle diameter increases, though this is offset
somewhat by the decrease in concentration with particle size above 50 nm.

It follows that aerosols incorrectly sized due to double and triple changing will be passed from the DMA
to the CCN and result in an additional uncertainty in the CCN measurements. To illustrate this, activated
fraction curves, were generated for 25, 50, 100, and 200 nm sodium chloride particle selection by the
DMA (Fig. 5).  The activation of sodium chloride is represented by sigmoid curves, where the midpoint
of each activation curve is the κ-Köhler-derived critical supersaturation of sodium chloride, and the
standard deviation of each curve is one-tenth of this value (consistent with the standard
deviation/midpoint ratio observed from our instrument's ammonium sulfate CCN calibration data).  For
each particle diameter, $D$, the observed activated fraction, $AF^{SS}_{D,weighted}$, for each percent supersaturation
$SS$ was determined by weighting the activated fraction $AF^{SS}_{D,i}$ of each particle diameter/charge at that
percent supersaturation, by the fraction of particles of that diameter:

$$AF^{SS}_{D,weighted} = \sum_{i=1}^{i=3} \frac{concentration\ of\ particles\ with\ charge\ i\ and\ diameter\ D}{concentration\ of\ particles\ with\ charge+1,+2,+3,and\ diameter\ D} AF^{SS}_{D,i} \qquad (15)$$


The raw data shown in Fig. 5 (green curves) can be corrected for multiple charging by determining the
fraction of particles with $> +1$ charge from the lower plateau in each plot (dashed lines). The adjusted
activated fraction for each percent supersaturation, $AF_{adjusted}$, is calculated using the equation:

$$AF_{adjusted} = \frac{AF_{raw} - AF_{plateau}}{1 - AF_{plateau}}$$
(16)


where $AF_{raw}$ is the raw activated fraction at that percent supersaturation, and $AF_{plateau}$ is the activated
fraction corresponding to the lower plateau (Rose, 2008). The adjusted activated fraction curves are
shown in Fig. 5 (blue curves). These are in good agreement with the theoretical κ-Köhler-derived
activation curves for sodium chloride (not shown).

Critical supersaturation was determined for each diameter by calculating the percent supersaturation at
which the raw $AF_{D,weighted}^{SS} = 0.5$. These critical supersaturations are shown in Fig. 6a, and the
theoretical critical supersaturations calculated from $\kappa$-Kohler theory are shown for comparison. Eq. 4
was used to calculate apparent hygroscopicity for each particle diameter, shown in Fig. 6b. A dashed
line in Fig. 6b indicates the literature value for $\kappa_{app}^{NaCl}$. It is apparent that failing to account for multiply-
charged particle in the activated fraction curves shown in Fig. 5 leads to an overestimation of $\kappa_{app}$ .
Artifacts in $\kappa_{app}$ are shown in Fig. 6c.

For the theoretical aerosol distribution used in this analysis (Fig. 3a), small, positive deviations from κ-
Köhler theory and the literature value for $\kappa_{app}^{NaCl}$ were observed ($0.01 \leq \kappa_{app,artifact}^{NaCl} \leq 0.04, 1 -$
3 % of $\kappa_{app}^{NaCl}$). As shown in the figure, $\kappa_{app}$ artifacts resulting from unaccounted-for multiple charges
decrease with particle diameter for this theoretical aerosol population. Greater $\kappa_{app}$ artifacts would be
expected for aerosol populations with more prevalent accumulation modes.

The aerosol/sheath ratio within the DMA also modulates the effect of multiple charges on $\kappa_{app}$. As the
aerosol/sheath ratio increases, the transfer function broadens, allowing particles that are both larger and
smaller than the selected diameter to exit the DMA. This in turn broadens the CCN activated fraction
curve (Rose et al., 2008). The larger particles will activate as CCN at lower supersaturations than
particles with the selected diameter, resulting in an increase in the activated fraction plateau due to
multiple-charged particles and a further decrease in the determined $SS_{crit}$. Petters et al. 2007b showed
that CCN activated fraction curves are significantly skewed by multiply-charged particles when the
mode diameter of the aerosol population upstream of the DMA exceeds the critical diameter of the size-
selected particles. In an example CCN activated fraction curve, Rose et al. 2008 demonstrated that a 1:6
ratio of doubly-to-singly charged particles resulted in an underestimation of the critical activation
diameter by 2%. Zhao-Ze and Liang, 2014 also showed that multiply-charged particles can introduce
significant uncertainty into hygroscopicity calculations.

**3.1.4    Additional artifacts resulting from DMA measurements**

Several additional factors that may impact experimental $\kappa_{app}$ are beyond the scope of this study, but are
worth mentioning as they represent additional potential sources of error in some cases. First, volatile
aerosols may partially evaporate inside the DMA, resulting in a decrease in particle size exiting the
DMA. DMA sizing error due to aerosol volatility (defined as the ratio of sampled diameter to the
selected diameter) increases with volatility, though sizing error can be decreased by increasing the
sheath flow rate in the DMA. Conversely, hygroscopic aerosols may grow inside the DMA, resulting in
larger particles existing the DMA. Operationally, errors in DMA sizing due to hygroscopic growth can
be mitigated if aerosols entering the DMA inlet are in wet metastable states (higher aerosol RH at DMA
inlet), and if DMA sheath flow rates are kept low (Khlystov, 2014).

Voltage shifts within the DMA (differences between the selected voltage and the actual voltage inside
the DMA) can lead to discrepancies between selected and sampled particle diameters.  Voltage shifts
may result from a space-charge field generated by the motion of charges within the DMA. Particles
charged by the bi-polar neutralizer will either be attracted towards or repelled away from the inner
column of the DMA, depending on whether they are positively or negatively charged.  This charge
separation creates a space-charge field which shifts the actual voltage within the DMA from the selected
voltage.  The impact of the space-charge field on the midpoint and spread of the DMA transfer function
increases as particle mobility increases (as particle size decreases), and as particle concentration
increases (Alonso and Kousaka, 1996; Alonso et al., 2000; Alonso et al., 2001).

**4. Artifacts derived from condensation particle counters**

**4.1 CPC operation at low concentration**

The second instrument which must function accurately during CCN experiments is the condensation

particle counter. CPC performance is characterized by the maximum counting efficiency (which may be

influenced by the working fluid in the instrument) and the 50 %-cut-off diameter ($d_{50}$), the particle

diameter at which 50 % counting efficiency is observed, both of which can vary between commercially

available models and even between individual CPCs (Heim et al., 2004). One study found that n-

butanol CPCs (TSI, Inc. Models 3772, 3775, and 3776) exhibited smaller $d_{50}$ for silver particles than

sodium chloride (3.3 $nm \leq d_{50}^{Ag} \leq 7.8\ nm$ and 4.1 $nm \leq d_{50}^{NaCl} \leq 14.7\ nm$), due to the more effective

condensation of n-butanol on silver particles (Hermann et al., 2007).

Maximum counting efficiencies in that study varied from 88.9 % to 100.3 %. Another comparison of n-

butanol CPCs (TSI Inc. Models 3010 and 3022, Grimm Tech. Inc. Model 5.403) found 3.1 $nm\ \leq d_{50} \leq$

11.9 $nm$ for sodium chloride aerosols (Heim et al., 2004). In another study, the counting efficiencies

observed in measurements of tungsten oxide particles by different instruments of the same model (TSI

3025) varied from 88.9 % to 138.9 %, while $d_{50}^{WO_x}$ varied from 3.2 nm to 11.0 nm (Hameri et al., 2002).

While some issues can cause undercounting at all concentrations, the additional issue of uncounted

particles due to the arrival of more than one particle in the detector's field of view at any time arises

only at higher concentrations. The cut-off between "low" and "high" concentration is not exact and

varies between instruments. CPC undercounting issues which arise even at relatively low concentrations

(which one would expect to encounter under standard experimental conditions) will be discussed in this

section.    Concentration-dependent effects encountered at higher concentrations will be explored in Sect.

471    4.2.


Six counting efficiency curves were generated using sigmoidal distributions and the 50 % cut-off
diameters and maximum counting efficiencies listed in Table 3.  Chosen values represent $d_{50}$ values and
maximum counting efficiencies reported in the literature under relatively low concentrations of 1000-
4000 cm$^{-3}$ (Hermann et al., 2007).  The resulting sigmoidal distributions (Fig. 7a) were used to
determine the counting efficiency of 25, 50, 100, and 200 nm particles.

Next, $\kappa_{app}$ was calculated from theoretical critical percent supersaturations for each chosen diameter. To
do so, four sigmoid curves representing sodium chloride CCN activation (hereon referred to as
"activation curves") for 25, 50, 100, and 200 nm were generated.  The κ-Köhler-$SS_{crit}$ of sodium
chloride was used as the midpoint of each activation curve, and one-tenth of this value was used as the
standard deviation (100 % CE, Fig. 7b-e).  These values are consistent with the standard
deviation/midpoint ratio observed from our instrument's ammonium sulfate CCN calibration data.

Activation curves were then generated for CPC Cases 1-6 by dividing the activated fraction for each dry
particle diameter by the counting efficiency for that diameter.  $SS_{crit}$ was determined for each CPC case
by finding the percent supersaturation at which activated fraction = 0.50.  Results are summarized in
Fig. 7f.  Next, critical supersaturation was converted to saturation, and $\kappa_{app,theory}$ was calculated for
each diameter in each CPC Case using Eq. (4) (see Fig. 7g). As above, $\kappa_{app}$ artifacts were calculated by
finding the difference between these results and the literature value of $\kappa_{app}$ for sodium chloride (see Fig.
7h).

For the diameters studied, the effect of maximum counting efficiency on CPC concentration (and
activated fraction) is greater than the effect of 50 %-cutoff diameter.  However, neither characteristic
resulted in large $\kappa_{app}$ artifacts.  The largest $\kappa_{app}$ artifact observed at "low" concentrations was 0.035 for
CPC Case 4, 2.4 % of the literature value for the apparent hygroscopicity factor for sodium chloride.

**4.2 CPC operation at high concentration**

Operation at high concentrations introduces an additional source of undercounting through particle coincidence at the CPC optical counter. For the TSI 3010 CPC, undercounting is observed is for particle concentrations above $1 \times 10^4 \ cm^{-3}$. At $5 \times 10^4 \ cm^{-3}$, the detector saturates and cannot detect higher concentrations. By comparison, the TSI 3025 is effective at counting higher particle concentrations, of up to $2.5 \times 10^4 \ cm^{-3}$ (Hameri et al., 2002;Sem, 2002).

To model undercounting due to particle coincidence, four CPC counting curves (Fig. 8a) were generated using the equations in Table 4. Case 7 represents a CPC where counting efficiency decreases with particle concentration, without reaching saturation. Cases 8-10 represent CPCs where saturation is reached at $4 \times 10^4 \ cm^{-3}$, $2 \times 10^4 \ cm^{-3}$, and $1 \times 10^4 \ cm^{-3}$, respectively. These saturation concentrations are of similar magnitude to those observed from TSI 3010 concentration data. It should be noted that the CPC concentration in Cases 7-10 levels off at the saturation concentration for each case.

In order to assess the importance of undercounting in CPC Cases 7-10, four theoretical aerosol distributions with a peak concentration at 50 nm were employed (Table 5, Fig. 8b). CPC Distribution 1 represents a worst-case scenario of similar magnitude to the highest particle concentrations measured during a coastal nucleation event (Hameri et al., 2002; Sem, 2002), while CPC Distributions 2, 3, and 4 are lower in concentration (due to the lack of undercounting in CPC Distributions 2, 3, and 4 as demonstrated in Figure 6b, the remaining analysis for CPC operation at high concentration considers only CPC Distribution 1.) CPC Cases 8-10 were applied to CPC Distribution 1 in order to determine the

concentration measured by the CPC for 25, 50, 100 and 200 nm aerosols.  The counting efficiency was
then calculated for each case and aerosol diameter in CPC Distribution 1.

Sigmoidal activated fraction curves were generated for 25, 50, 100 and 200 nm sodium chloride
aerosols.  As in the low concentration cases, the midpoint of each 100 % CE curve was chosen to be
equal to the κ-Köhler-derived $SS_{crit}$ of sodium chloride at each dry diameter, and the standard deviation
of each curve is equal to one-tenth of the $SS_{crit}$.  These activated fraction curves were adjusted using the
counting efficiencies calculated in the previous step.  In cases where the activated fraction has increased
due to undercounting by the CPC, the theoretical sigmoidal curve shifts to the left relative to the 100 %
CE case (Fig. 8c-f).  Thus, undercounting by the CPC effectively increases the reported activated
fraction. As before, $SS_{crit}$ was determined from each of these curves, and $\kappa_{app,theory}$ was subsequently
calculated using Eq. (4) (Fig. 8g-h).

$\kappa_{app,theory}$ fell over a much wider range for 25, 50, and 100 nm particles (1.30-1.56, 1.32-1.70, and
1.30-1.55, respectively) than for 200 nm particles (1.28-1.29) due to the lower concentration of 200 nm
particles in the chosen aerosol distribution, which resulted in a higher counting efficiency for these
aerosols.  In comparison, the largest range in $\kappa_{app,theory}$ was observed for 50 nm aerosols, the peak
diameter in this aerosol distribution.

A wider range in $\kappa_{app,theory}$ was observed for the high-concentration CPC Cases (7-10) compared to the
low-concentration CPC Cases (1-6).  The lowest counting efficiency observed across the low-
concentration cases was 89.9 % for 25 nm aerosol in Case 4, while the lowest counting efficiency
observed in the high-concentration cases was 18.0 % for 50 nm aerosol in Case 10.

Artifacts in the apparent hygroscopicity parameter are shown in Fig. 8i. $\kappa_{app}$ artifacts were the greatest
for a CPC that becomes saturated at 20,000 particles/cm$^3$ ($0.0131 \leq \kappa_{app} \leq 0.4206$). The lower the
concentration at which a CPC becomes saturated, the more quickly its counting efficiency will drop as
concentration increases, resulting in increased activated fraction and increased apparent hygroscopicity.
The magnitude of artifacts due to CPC undercounting depends on the saturation concentration of the
CPC and the distribution of the aerosol population being studied.

## 5. Artifacts derived from cloud condensation nuclei instruments

Finally, the third instrument whose performance accuracy contributes to the overall certainty in CCN assessment in the CCN instrument itself. Several instruments have been implemented for measuring CCN concentrations over the last few decades. Older models include the Continuous Flow Parallel Plate Diffusion Chamber (Sinnarwalla, 1973) and the Hudson CCN spectrometer (Hudson, 1989) which both employ an applied temperature gradient perpendicular to the aerosol flow. Newer models, such as the widely-used Droplet Measurement Technology Cloud Condensation Nuclei Counter (DMT CCN-100), operate with a streamwise temperature gradient and continuous, laminar flow (Lance et al., 2006). The total flow through the DMT CCN-100 is 0.20-0.90 L min$^{-1}$, though the instrument is typically operated with a total flow of 0.50 L min$^{-1}$. The aerosol/sheath ratio in the DMT CCN-100 is set by the user, and a ratio of 1:10 is commonly chosen. The following analysis considers the DMT CCN-100. According to the CCN-100 manual, the counting efficiency for this CCN instrument depends on concentration and supersaturation (Fig. 9a). The counting efficiency decreases rapidly with concentration at < 0.2 % SS due to rapid water vapor depletion at these low supersaturations, and falls off more slowly for > 0.2 % SS (DMT CCN-100 manual).

The counting efficiency of the DMT CCN-100 was tested for four lognormal aerosol distributions with peak concentrations at 50 nm and varying total concentrations (Table 5, Fig. 9b). Note that CCN Cases 1-4 are identical to the aerosol distributions CPC Distributions 1-4 used for the high-concentration CPC cases.

The counting efficiencies for each case were applied to theoretical sodium chloride sigmoidal activated fraction curves to produce normalized activated fraction curves (Fig. 9c-f). As above, the midpoint is

set to the $SS_{crit}$ of sodium chloride at each dry diameter, and the standard deviation is assumed to be
one-tenth of $SS_{crit}$. CCN undercounting effectively decreases activated fraction, therefore shifting the
activated fraction curve downwards and towards higher percent supersaturations. The opposite effect is
observed when CPC undercounting occurs. Critical supersaturation was determined for each CCN case,
as above (Fig. 9g). Values of $SS_{crit}$ were then converted to saturation, and $\kappa_{app,theory}$ was calculated
using Eq. (4) (Fig. 9h).

Significant deviations from κ-Köhler theory were only observed in CCN Case 1, with total aerosol
concentration $5 \times 10^6$ particles/cm³ (Fig. 9g-i). The largest deviation for CCN Case 1 was observed in
100 nm particles ($\kappa_{app,artifact} = -0.57$), due to the higher concentration of 100 nm particles compared
to 25 and 200 nm particles, and the lower percent supersaturation necessary for activation. The largest
artifacts across CCN Cases 2 and 3 were also observed for 100 nm particles, though no artifacts were
observed for any particle diameter in CCN Case 4 due to the much lower concentrations.

Sodium chloride is very hygroscopic. It should be noted that aerosols consisting of less hygroscopic
compounds will activate at higher percent supersaturations (> 0.2 % SS regime) which will lead to
smaller $\kappa_{app}$ artifacts when the same aerosol distribution and total aerosol concentration is considered.
If a mixture was considered (for example, sodium chloride with a non-hygroscopic species such as soot)
the results may also be different. The shape of the aerosol distribution must also be taken into account.
A distribution with a narrower peak than the one generated for this analysis would be at risk for larger
$\kappa_{app}$ artifacts for any total aerosol concentration, and these artifacts would be greater at the peak
diameter, while a broader distribution would result in less variation in $\kappa_{app}$ artifacts for each particle
diameter.
6. **Counting statistics in CCN and CPC measurements**

Though it is beyond the scope of this analysis, it should be mentioned that sampling at very low particle
concentrations ($< 200$ cm$^{-3}$ total particle concentration) can introduce additional error into CCN and
CPC measurements.  This error can be mitigated by increasing scan times (Moore et al., 2010).  For
example, Moore et al., 2010 averaged CCN and particle concentrations over 5-second intervals for
monodisperse particle concentrations $< 10$ cm$^{-3}$, and increased averaging time to 20-second intervals
when the monodisperse particle concentration reached $< 6$ cm$^{-3}$.

**7.  Discussion**

A comparison of the major instrument sources of error in CCN-derived $\kappa_{app}$ is shown in Fig. 10. In
addition, the best and worst case combination of errors, determined by additive error propagation, are
also shown.  DMA Case 4, CPC Case 4, CPC Case 10, and CCN Case 1 represent the operating
conditions that resulted in the largest $\kappa_{app}$ artifacts in this study.  In DMA Case 4, the aerosol/sheath
ratio of 0.30 resulted in a broadened aerosol distribution downstream of the DMA.  Compared to DMA
Case 1, where $Q_a/Q_{sh} = 0.10$, the downstream diameter range in DMA Case 4 was 300 % higher for 25
nm particles, resulting in a spread of 20-36 nm.   Similarly, the diameter ranges for 50, 100, and 200 nm
diameter were 220 %, 230 %, and 220 % wider than in Case 1, respectively.  Compared to the most ideal
DMA case presented in this study (DMA Case 2), where $Q_a/Q_{sh} = 0.05$, the downstream diameter
range in DMA Case 4 was 700 % higher for 25 nm particles; the diameter ranges for 50, 100, and 200
nm diameter were 540 %, 560 %, and 520 % wider than in Case 2, respectively.   The results
demonstrate that limiting $Q_a/Q_{sh}$ to $\leq 0.10$ will result in a narrow particle size distribution downstream
of the DMA.  Other studies have recommended employing DMA sample/sheath ratios of 0.2 (Petters et
al., 2007; Carrico et al., 2008; Moore et al., 2010) or 0.1 (Moore et al., 2010; Zhao-Ze and Liang, 2014)
in order to minimize measurement aerosols due to transfer function broadening.

The effects of multiply-charged particles on $\kappa_{app}$ calculations were also quantified, as shown in Fig. 10.
Small, positive $\kappa_{app}$ artifacts $(1 - 3 \text{ % of } \kappa_{app}^{NaCl})$ were observed when particles with +2 and +3 charges
were not accounted for. This analysis considered a theoretical aerosol distribution in which most of the
particles measure less than 100 nm in diameter. Actual aerosol distributions vary temporally and
spatially, and often include accumulation and coarse modes that would result in larger $\kappa_{app}$ artifacts.

CPC Case 4 represents $\kappa_{app}$ artifacts (0.031-0.035) due to undercounting that arises from poor
maximum CPC counting efficiency (90 %), which may be observed when using butanol as a working
fluid while measuring the concentration of inorganic aerosols. In contrast, $\kappa_{app}$ artifacts are negligible
$(< 0.10 \text{ % of } \kappa_{app}^{NaCl})$ in CPC Case 3, where maximum counting efficiency = 100 %. CPC Cases 8 and 10
(applied to the highest-concentration case, CPC Distribution 1) represent undercounting at high
concentration with CPCs where saturation is observed at $4 \times 10^4 \ cm^{-3}$ and $1 \times 10^4 \ cm^{-3}$, respectively.
Counting efficiency drops off more rapidly with concentration in the latter case, resulting in $\kappa_{app}$
artifacts that are highest at the peak of the aerosol distribution (0.1190 and 0.4206 for 50 nm aerosols in
CPC Cases 8 and 10, respectively). It should be noted that undercounting was only observed for one of
the four upstream distributions studied, CPC Distribution 1. No undercounting was observed when CPC
Cases 7-10 were applied to CPC Distributions 2-4.

CCN Case 1 represents CCN undercounting at high concentration (total aerosol concentration = $5 \times$
$10^6 cm^{-3}$). CCN undercounting is greatest for low supersaturation (< 0.2 %) and high concentration,
resulting in the lowest counting efficiency and highest $\kappa_{app}$ artifacts (- 0.57) for 100 nm aerosols in
CCN Case 1. The largest CCN-derived $\kappa_{app}$ artifact observed outside of CCN Case 1 was $-$ 0.01 for
100 nm aerosols in CCN Case 2.

The combined artifacts for the cases where the highest artifacts were observed (DMA Case 4, multiple
particle charging, CPC Case 4, CPC Case 10, CCN Case 1) are 0.24, 0.21, 0.23, and 0.15 for 25, 50,
100, and 200 nm particles respectively, as shown in Fig. 10. The combined artifacts for the lowest-
artifact cases (DMA Case 2, CPC Case 3, and CCN Case 4) are < 0.008 except for all four particle
diameters.


**Conclusions**

The sensitivity of weather and climate models to accuracy in CCN activation predictions has been demonstrated in other works. Possible sources of apparent hygroscopicity artifacts calculated from CCN measurements have been presented in this study. This analysis has focused on sodium chloride and ammonium sulfate aerosols, but it can be extended to other aerosol populations, including mixtures and field samples.

The greatest combined artifacts ($0.15 < \kappa_{app,artifact} < 0.24$, NaCl) occurred as a result of the combined issues of the highest DMA aerosol/sheath ratio, uncorrected multiple particle charging, and undercounting by both CPC and CCN instrument. The lowest combined artifacts ($0.0021 < \kappa_{app,artifact} < 0.0074$, NaCl) occurred as a result of ideal operating conditions: lowest DMA/sheath ratio, corrected multiple particle charging, and little to no undercounting.

The largest single-instrument artifacts ($-0.57 < \kappa_{app,artifact} < 0.42$ for sodium chloride) in this study arise from undercounting by either the CPC or CCN counter at high concentration. This problem occurs during attempts to measure aerosol concentrations of $\sim 10^4$ cm$^{-3}$ which is much higher than the recommended concentration ranges for either instrument, (CPC Cases 7-10 and CCN Case 4). Corrective action should be taken to dilute aerosol samples in order to avoid undercounting. It should be noted that these artifacts are for individual instruments and do not take combined operation of the CPC and CCN into account; when both instruments undercount, artifacts in $\kappa_{app,artifact}$ are reduced.

Smaller single-instrument artifacts ($\kappa_{app,artifact} < 0.04$) were observed for the CPC cases where 50 %-cut-off diameter and maximum counting efficiency were varied. Given the chosen particle diameters

(25, 50, 100, 200 nm), $\kappa_{app}$ artifacts due to $d_{50}$ were minimal.  The largest $\kappa_{app}$ artifacts for a CPC
counting at low concentration (0.031-0.035) were observed where the maximum counting efficiency was
equal 0.90.  This may represent a compositional mismatch between n-butanol as the working fluid and
sodium chloride as the aerosol, due to the poor solubility of the latter in the former.  Individual n-butanol
CPCs may exhibit higher maximum counting efficiencies for sodium chloride.

Uncertainty arising from the DMA depended greatly on the chosen aerosol and sheath settings. One set
of DMA cases (Cases 2-4) examined the effect of aerosol/sheath ratio.  By decreasing this ratio, a
narrower near-monodisperse flow can be produced, which increases the accuracy of calculated $\kappa_{app}$.
The $\kappa_{app}$ artifacts for an aerosol/sheath ratio of 0.10 were 0.65 % of $\kappa_{literature}$ for 25 nm sodium
chloride aerosols, 0.31 % for 50 nm, -0.17 % for 100 nm, and -1.2 % for 200 nm.

The second set of DMA cases (5-7) were designed to evaluate the effects of holding the sheath flow
constant while varying the excess air flow by -2 %, +2 %, and +5 %. These resulted in shifts of $\leq$ 2 nm
for 25 nm and 50 nm particles, $\leq$ 4 nm for 100 nm particles, and $\leq$ 7 nm for 200 nm particles.  The
downstream aerosol distribution was shifted towards larger particle diameters when sheath flow
exceeded excess flow, and towards smaller particle diameter when sheath flow was less than excess
flow.  When taking field measurements, the composition of the sample may vary with particle diameter,
thereby introducing another source of error from a broader DMA distribution.
By extension, the issue of uncertain sizing by the DMA leads to added uncertainties in the CCN
instrument calibrations which are strongly dependent on the chosen aerosol to sheath ration within the
DMA.  We recommend conducting all CCN calibrations with DMA aerosol to sheath ratio of 1:10 or
1:20 which will reduce kappa uncertainties to less than 1% for all dry sizes (25 to 200 nm).

Overall, under optimal operating conditions, where the DMA aerosol/sheath ratio is 0.10 and
excess/sheath ratio is 1.0, and in the absence of undercounting by the CPC or CCN, uncertainties in $\kappa_{app}$
are less than ±1.2 % for 25 to 200 nm particles. During sampling, when the DMA sample/sheath ratio is
reduced to 0.05, $\kappa_{app}$ uncertainties decrease to ±0.58 %.  Additionally, errors in activated fraction (and
therefore $\kappa_{app}$) resulting from the bipolar charge distribution can be corrected by determining the
fraction of particles with multiple charges.

In this study, apparent hygroscopicity parameter artifacts were calculated for two pure, inorganic species
in this study.  This analysis could be used to estimate $\kappa_{app}$ artifacts for ambient aerosol populations,
which may result in a better understanding of the "real' differences between these populations.  As
discussed in the introduction, Collins et al. 2016 aggregated $\kappa_{app}$ from several mesocosm and field
studies for 30-100 nm sea spray aerosol ($0.4 < \kappa_{app}^{SSA} < 1.3$).  The wide range of $\kappa_{app}$ in these studies
may be attributed to differences in composition, experimental artifacts, or a combination of the two.
Quantification of experimental artifacts would facilitate interpretation of $\kappa_{app}$ in aerosol populations and
constrain the importance of composition in CCN activation. There has been a recent proliferation of
CCN data availability from multiple researchers and multiple experimental setups.  To maximize the
utility of these studies and to compare cloud-activating properties of various ambient aerosol masses, it
is essential that artifacts are considered in both CCN data collection and in reporting of the data.
**Supplement Link**
Will be included by Copernicus

**Author Contribution**
Sarah D. Brooks provided the conceptual framework and contributed to the writing of the manuscript.
Jessica A. Mirrielees performed the analysis and lead the writing of the manuscript.

**Competing Interests**
The authors declare that they have no conflict of interest.

**Disclaimer**
Will be included by Copernicus

**Acknowledgements**
This project was supported by the National Science Foundation of the United States (Award
#15398810).  In addition, Mirrielees thanks Texas A&M University for support through Institute for and
Advanced Studies HEEP PhD Fellowship and a Lechner Scholarship.

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

| Notation | |
|---|---|
| $\alpha_{CC}, \beta_{CC}, \gamma_{CC}$ | Empirically-determined constants used to calculate Cunningham slip correction factor |
| $Z_p$ | Aerosol particle electrical mobility |
| $C_C$ | Cunningham slip correction factor |
| $d_m$ | Electrical mobility diameter |
| $n$ | Number of charges on particle |
| $e$ | Elementary unit of charge |
| $\eta$ | Gas dynamic viscosity |
| $\lambda$ | Mean free path |
| $Q_{sh}$ | Sheath flow |
| $Q_e$ | Excess air flow |
| $Q_a$ | Aerosol flow |
| $Q_s$ | Sample flow |
| $\kappa_{app}$ | Apparent hygroscopicity parameter |
| $\kappa_{app,artifact}$ | Apparent hygroscopicity parameter artifact |
| $s$ | Equilibrium water vapor saturation |
| $s_{crit}$ | Critical saturation (50 % of aerosols active as cloud condensation nuclei) |
| $A$ | Constant used in calculating $\kappa_{app}$ |
| $\sigma_{lv}$ | Surface tension of water |
| $T$ | Temperature |
| $D_{act}$ | Activation diameter |
| $SS_{crit}$ | Critical percent supersaturation |
| $\alpha_{TF}$ | Height of DMA transfer function |
| $\beta_{TF}$ | Half-width of DMA transfer function |
| $Z'_p$ | Mobility of particle at DMA inlet |
| $Z_{p,mid}$ | Midpoint of transfer function |
| $\Delta Z_p$ | Half-width of transfer function |
| $V_0$ | Voltage selected at DMA |
| $r_1$ | DMA inner radius |
| $r_2$ | DMA outer radius |
| $L$ | DMA length |
| $d_{50}$ | 50 %-cut-off diameter |


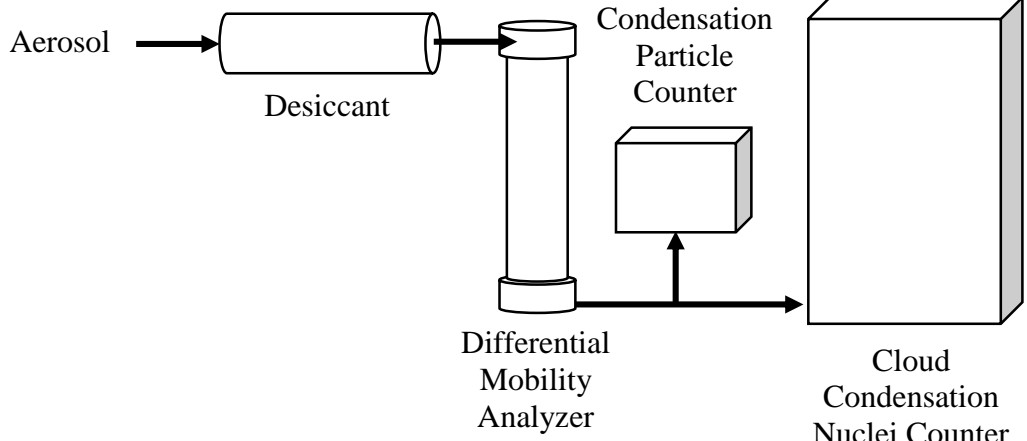


**Figure 1** Experimental setup used for obtaining sized CCN and particle concentration measurements
from an aerosol sample.

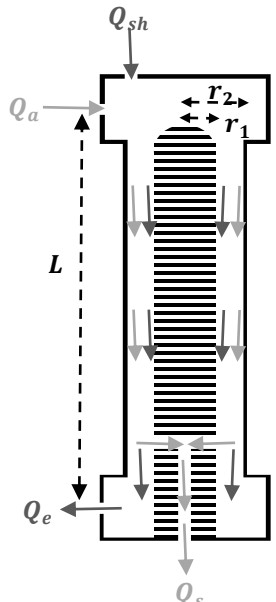


**Figure 2** Simplified flow diagram of a DMA with an inner electrode radius $r_1$, outer electrode radius $r_2$,
distance between aerosol inlet and sample outlet $L$, clean sheath air flow $Q_{sh}$, aerosol flow $Q_a$, excess air
flow $Q_e$, and sample air flow $Q_s$.

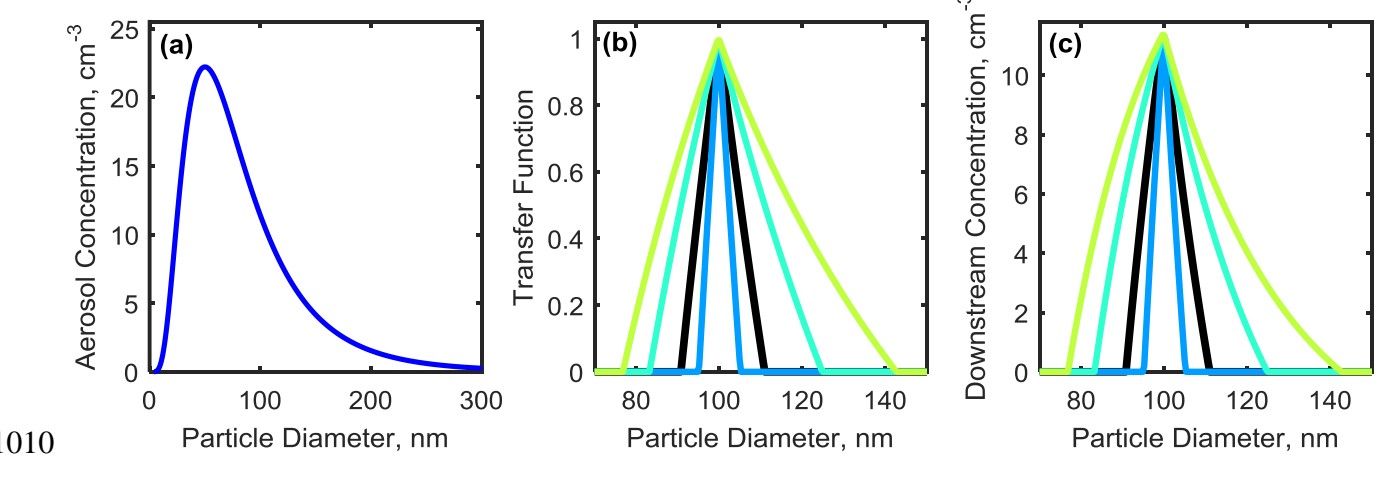



**Figure 3** (a) A theoretical aerosol distribution generated using a lognormal function centered at 50 nm. .
(b) The transfer function calculated using Eq. (7). (c) Multiplying the distribution by the transfer function
gives the downstream aerosol concentration ($cm^{-3}$).

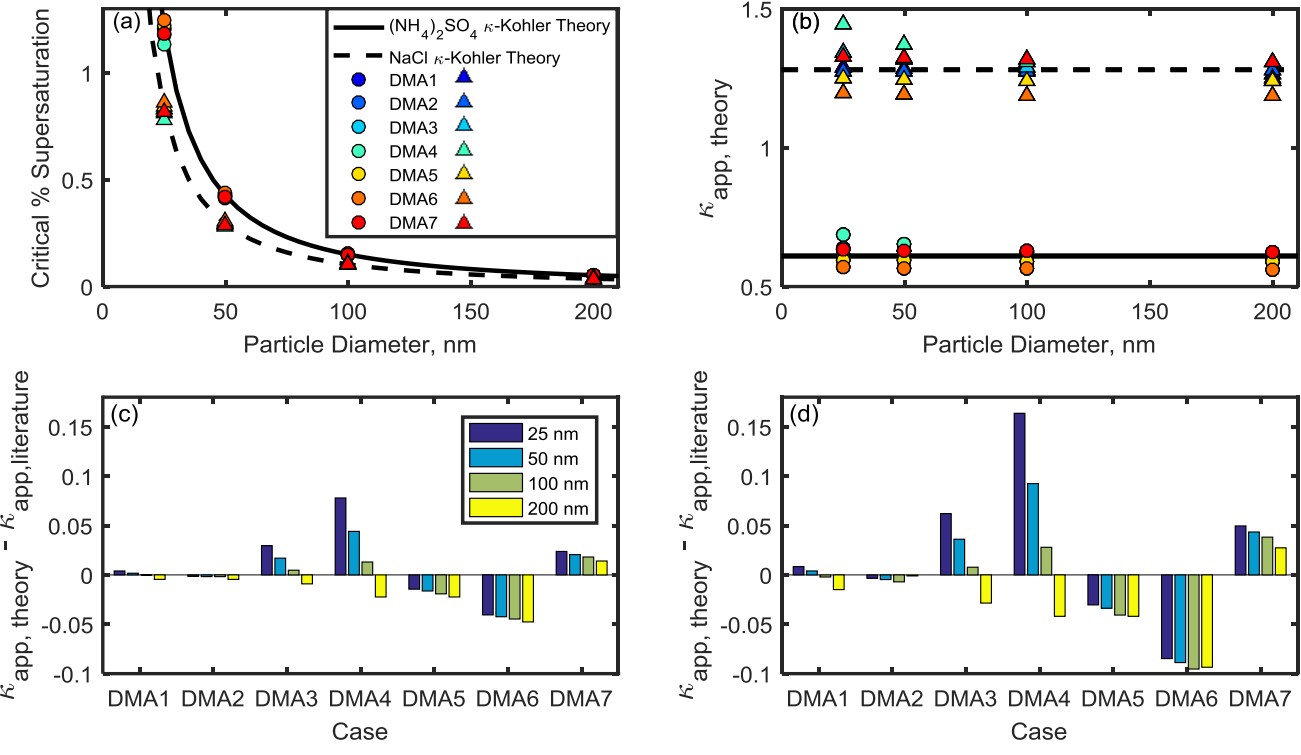



**Figure 4** (a) Critical supersaturation of ammonium sulfate and sodium chloride particles calculated for DMA Cases 1-7 for sodium chloride (triangles) and ammonium sulfate (circles). Ammonium sulfate and sodium chloride curves from $\kappa$-Köhler theory are shown for comparison. (b) Apparent hygroscopicity $\kappa_{app}$ for DMA cases 1-7. (c) DMA-flow-derived artifacts in ammonium sulfate $\kappa_{app}$ are shown for each DMA case. (d) DMA-flow-derived artifacts in sodium chloride $\kappa_{app}$ are shown for each DMA case.

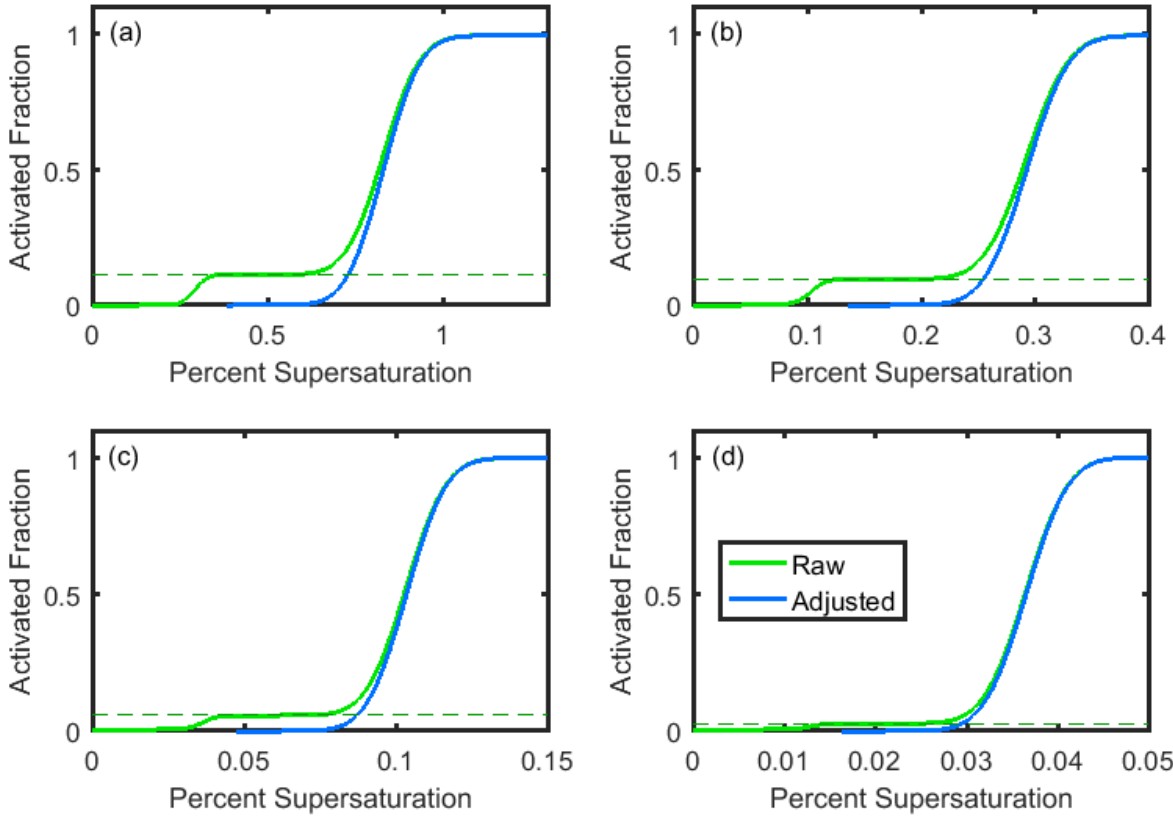


**Figure 5** Theoretical raw (green) and adjusted (blue) activated fraction curves for (a) 25 nm (+1), 50 nm
(+2), and 75 nm (+3) particles; (b) 50 nm (+1), 100 nm (+2), and 150 nm (+3) particles; (c) 100 nm
(+1), 200 nm (+2), and 300 nm (+3) particles; (d) 200 nm (+1), 400 nm (+2), and 600 nm (+3) particles.
All particles are pure sodium chloride.

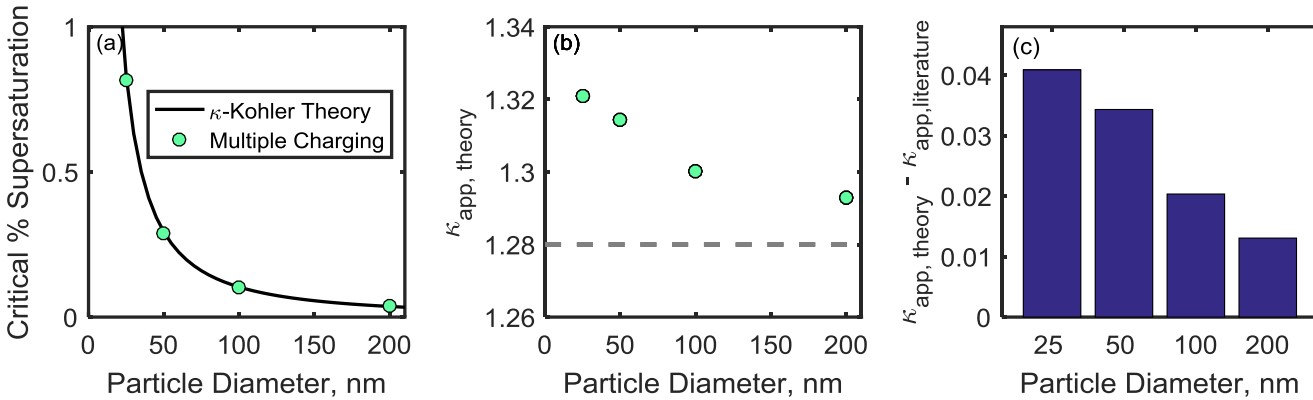


**Figure 6** (a) Critical percent supersaturation of sodium chloride particles determined from activated

fraction curves shown in Fig. 5. A κ-Köhler curve for sodium chloride is shown for comparison. (b)

Theoretical $\kappa_{app}$ for each particle diameter (gray dashed line indicates literature value for $\kappa_{app}^{NaCl}$). (c)

Artifacts in $\kappa_{app}$ resulting from multiple particle charges.

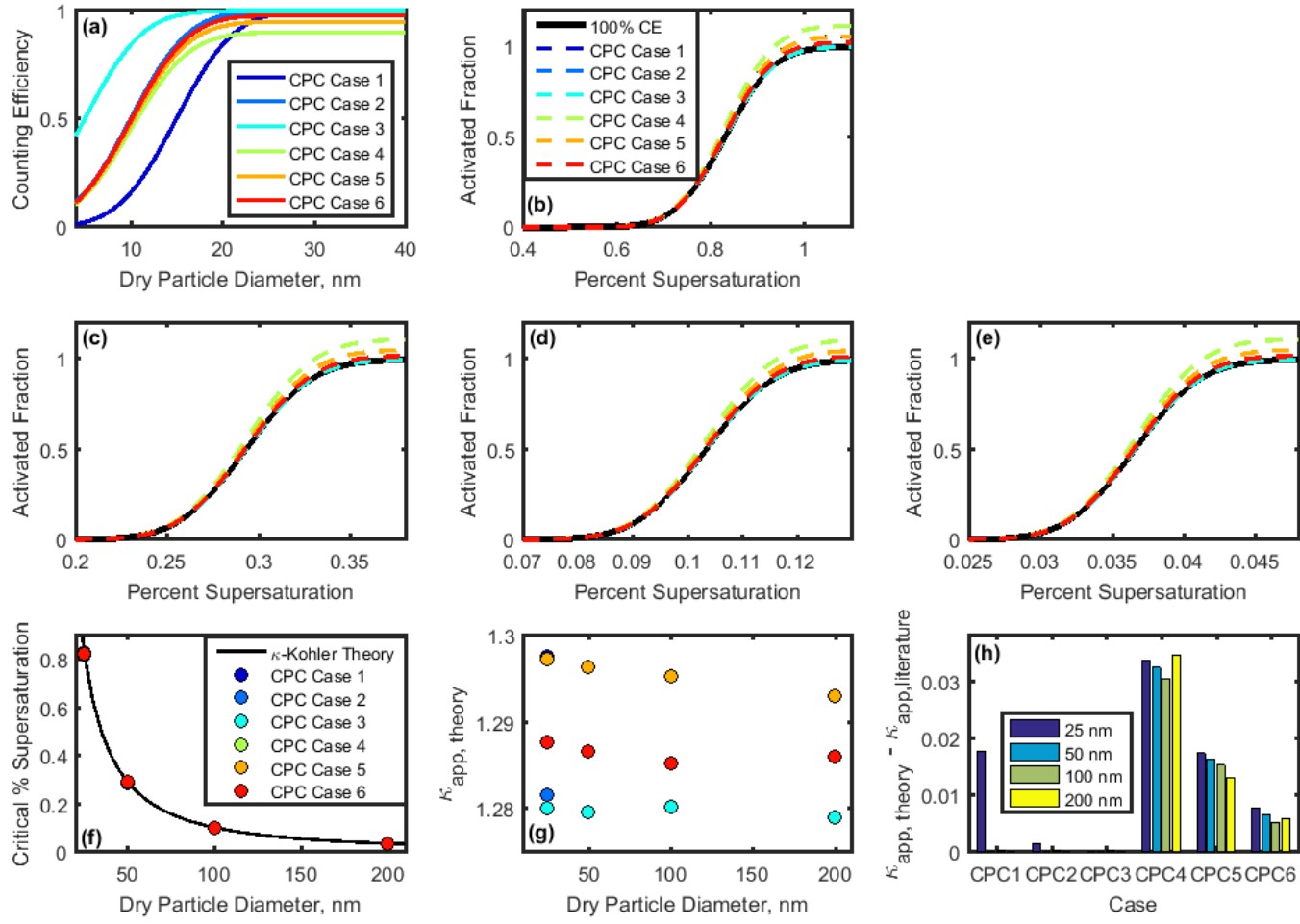

1032

**Figure 7** (a) Counting efficiency curves for CPC Cases 1-6 (shown in Table 3).

(b-e) CCN activated fraction curves for 25, 50, 100, and 200 nm NaCl, respectively. (f) Critical

supersaturation calculated for each particle diameter. (g) Theoretical $\kappa_{app}$ for each CPC case and

particle diameter. (h) Artifacts in $\kappa_{app}$ for each CPC case and particle diameter.

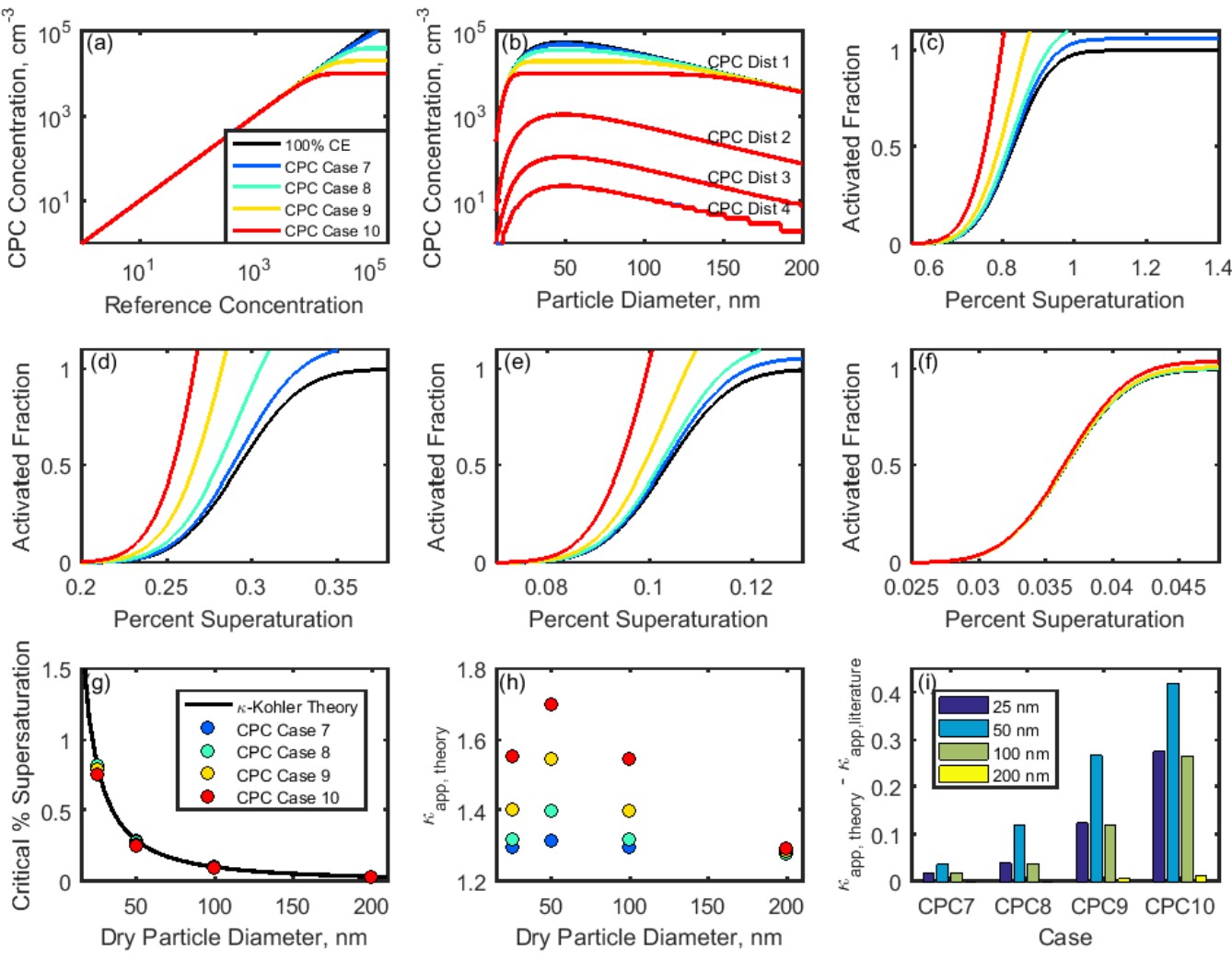

1037

**Figure 8** (a) Theoretical relationships between the reference aerosol concentration and CPC

concentration. (b) Concentration-dependent counting efficiencies from (a) were applied to four

theoretical aerosol distributions. (c-f) Activated fraction curves for CPC Distribution 1 and particle

diameters 25, 50, 100, and 200 nm NaCl aerosol, respectively. (g,h) Critical supersaturation and $\kappa_{app}$ for

each case. (i) Artifacts in $\kappa_{app}$ for each case.

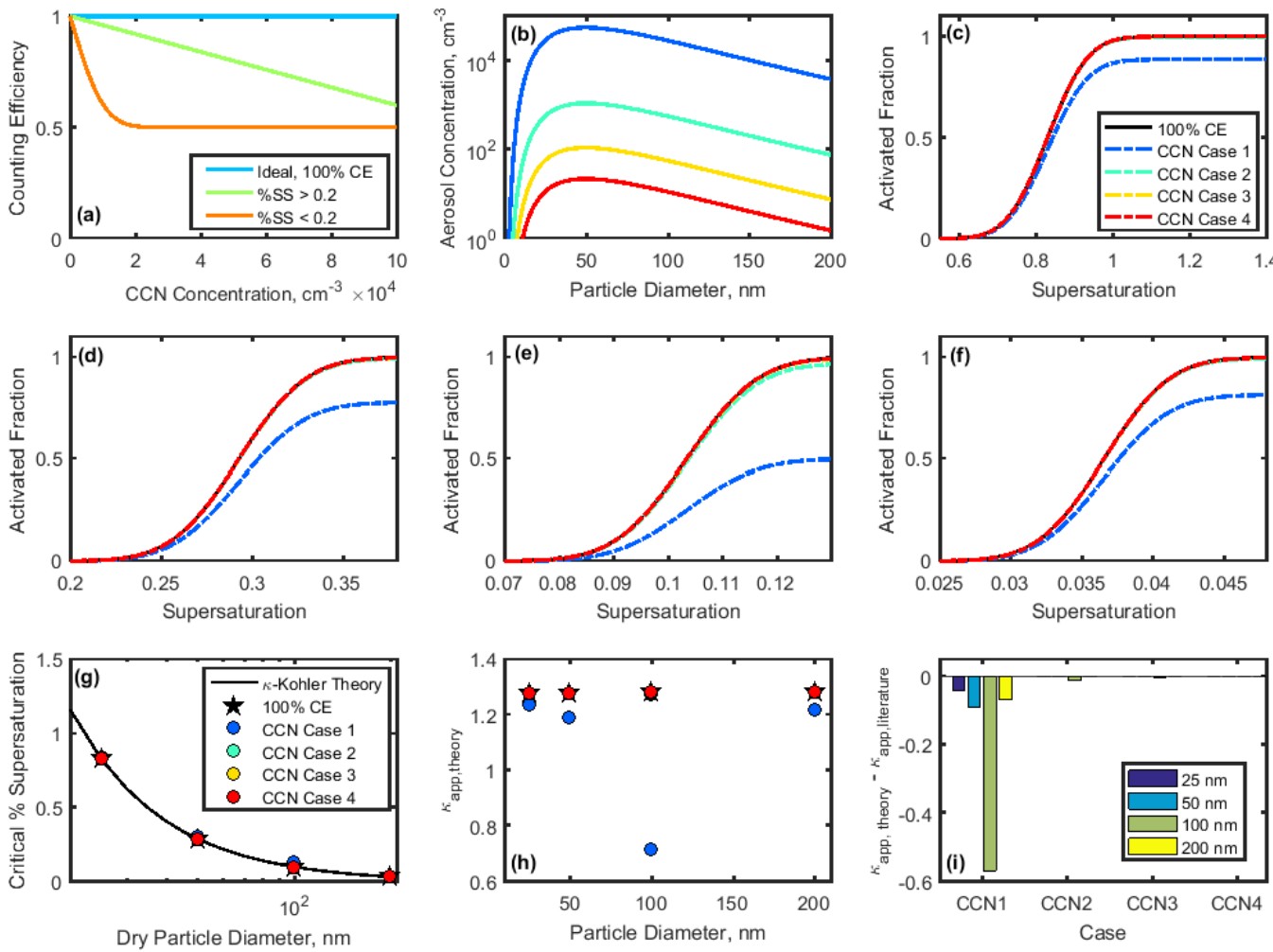


**Figure 9** (a) Counting efficiencies of the DMT CCN-100 for specific supersaturations. (b) Lognormal
aerosol distributions used to study CCN undercounting at high concentrations. (c-f) Activated fraction
curves for 25, 50, 100, and 200 nm NaCl particles. Supersaturation-specific counting efficiencies from
(a) applied to theoretical sigmoid curves for NaCl CCN activation. Activated fraction in the case of 100
% counting efficiency is shown for comparison. (g) Critical supersaturation for each case. (h)
Theoretical $\kappa_{app}$ calculated for each case. (i) Artifacts in $\kappa_{app}$ artifacts for each case.

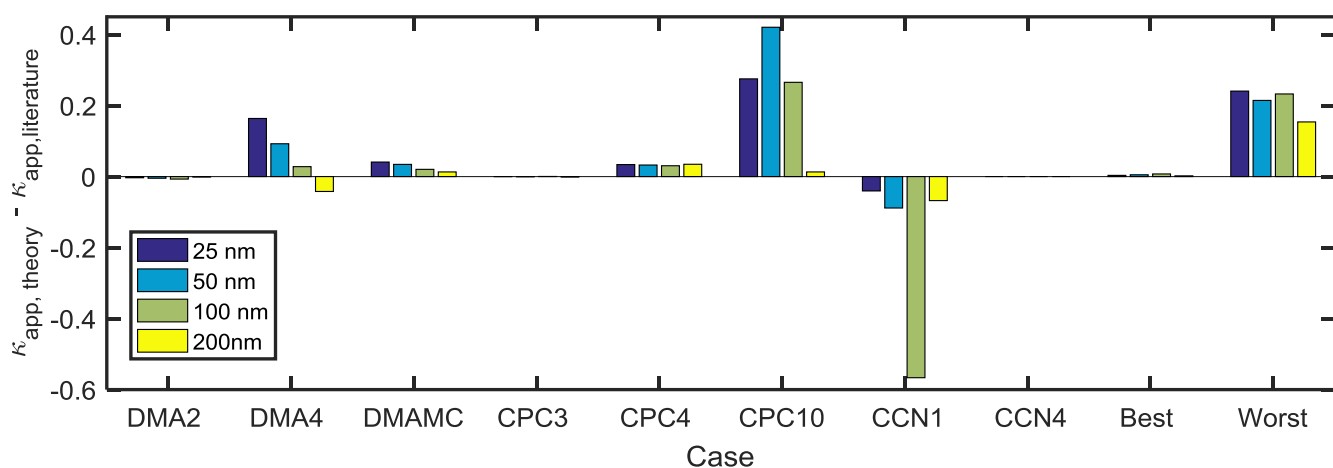


**Figure 10** Comparison of $\kappa_{app}$ artifacts derived from best and worst case scenarios for instrumental
measurements for sodium chloride. Combined artifacts for the lowest-artifact cases (Best: DMA Case
2, CPC Case 3, and CCN Case 4) and the highest-artifact cases (Worst: DMA Case 4, multiple charging,
CPC Case 4, CPC Case 8, and CCN Case 1).

**Table 1 Theoretical DMA Flow Test Cases**

| Case | $Q_{sh}$ (L min$^{-1}$) | $Q_e$ (L min$^{-1}$) | $Q_a$ (L min$^{-1}$) | $Q_s$ (L min$^{-1}$) | $Q_a/Q_{sh}$ | $Q_e/Q_{sh}$ |
|---|---|---|---|---|---|---|
| DMA 1 | 3.00 | 3.00 | 0.30 | 0.30 | 0.10 | 1.00 |
| DMA 2 | 3.00 | 3.00 | 0.15 | 0.15 | 0.05 | 1.00 |
| DMA 3 | 3.00 | 3.00 | 0.60 | 0.60 | 0.20 | 1.00 |
| DMA 4 | 3.00 | 3.00 | 0.90 | 0.90 | 0.30 | 1.00 |
| DMA 5 | 3.00 | 3.06 | 0.36 | 0.30 | 0.12 | 1.02 |
| DMA 6 | 3.00 | 3.15 | 0.45 | 0.30 | 0.15 | 1.05 |
| DMA 7 | 3.00 | 2.94 | 0.24 | 0.30 | 0.08 | 0.98 |


**Table 2 Predicted downstream particle diameter range for each DMA case.**

| Case | 25 nm | 50 nm | 100 nm | 200 nm |
|---|---|---|---|---|
| **DMA 1** | 23-27 | 46-56 | 91-111 | 181-222 |
| **DMA 2** | 24-26 | 48-53 | 95-105 | 190-211 |
| **DMA 3** | 21-31 | 42-62 | 83-125 | 167-250 |
| **DMA 4** | 20-36 | 39-71 | 77-143 | 154-285 |
| **DMA 5** | 23-27 | 45-55 | 90-110 | 181-220 |
| **DMA 6** | 22-27 | 45-54 | 89-107 | 178-215 |
| **DMA 7** | 23-28 | 46-56 | 92-112 | 183-225 |


**Table 3 Values of 50%-cutoff diameter and maximum counting efficiency used in investigating $\kappa_{app}$ artifacts for low particle concentrations measured by a CPC.**

| Case | $d_{50}$, nm | Maximum Counting Efficiency |
|---|---|---|
| CPC 1 | 15 | 100 % |
| CPC 2 | 10 | 100 % |
| CPC 3 | 5 | 100 % |
| CPC 4 | 10 | 90 % |
| CPC 5 | 10 | 95 % |
| CPC 6 | 10 | 98 % |


**Table 4 Equations used to model the relationship between a reference or "true" aerosol concentration $x$ (particles cm$^{-3}$), and the concentration measured by a condensation particle counter $y$ (particles cm$^{-3}$).**

| Case | Equation |
|---|---|
| CPC 7 | $y = x - 2 \times 10^{-6} x^2$ |
| CPC 8 | $y = 40000 \, \mathrm{erf}\left(\dfrac{x}{32000\sqrt{2}}\right)$ |
| CPC 9 | $y = 20000 \, \mathrm{erf}\left(\dfrac{x}{16000\sqrt{2}}\right)$ |
| CPC 10 | $y = 10000 \, \mathrm{erf}\left(\dfrac{x}{8000\sqrt{2}}\right)$ |


**Table 5 Total concentrations used in theoretical aerosol distribution for CPC operation at high concentration and CCN-derived $\kappa_{app}$ artifacts.**

| CPC Distribution | CCN Case | Total Concentration (particles cm⁻³) |
|---|---|---|
| CPC Distribution 1 | CCN 1 | $5 \times 10^6$ |
| CPC Distribution 2 | CCN 2 | $1 \times 10^5$ |
| CPC Distribution 3 | CCN 3 | $1 \times 10^4$ |
| CPC Distribution 4 | CCN 4 | $2 \times 10^3$ |
