# Peer review of "Instrument Artifacts Lead to Uncertainties in Parameterizations of Cloud"

_Atmospheric Measurement Techniques, 2018_

## Referee Comment (RC1) · Anonymous Referee #1 · 16 Jul 2018

The manuscript "Instrument Artifacts Lead to Uncertainties in Parameterizations of Cloud Condensation Nucleation" assessed the contributions of potential artifacts in CCN activation measurement to the uncertainties in hygroscopic parameter, kappa, using a theoretical way. The artifacts include the potential artifacts from DMA, CPC counting, and CCN counting. It presents results of several scenarios including various operating conditions of DMA (ratio of aerosol to sheath flow, ratio of sheath to excess flow), counting efficiency of CPC and CCN at varied aerosol concentrations. The authors found that the broadening of aerosol size distribution out of DMA by increasing the aerosol to sheath flow ratio led to an overestimate of kappa. The undercounting of particles by CPC at high aerosol concentrations led to an overestimate of kappa, while the undercounting of particles by CCN led to an underestimate of kappa.

Assessing the contribution of potential artifacts in CCN operation to the kappa is beneficial to the CCN community. The manuscript fits well the scope of AMT. However, I have some concerns before the manuscript is considered to be published in AMT.

General comments

1. This study used a pure theoretical approach to assess the artifacts in various CCN operating scenarios. However, many scenarios are not common in the real CCN activation measurement. For example, is very rare that the particle number concentrations at the output of DMA reach 1e4 # cm$^{-3}$, or even 5e6 # cm$^{-3}$ as investigated in the section of artifacts derived from CPC and CCN. The authors suggested in the introduction section that the discrepancy in experimental results for ammonium nitrate and some organics in the literature are contributed by the artifacts in CCN measurement. An interesting question is to what extent the artifacts investigated here can explain the discrepancies in the kappa of ammonium nitrate, for example, in the literature.
2. The approach used to derive artifacts from DMA in this study is significantly different from the real CCN measurement. Firstly, in the real CCN measurement, uncertainties in $SS_{crit}$ (accordingly kappa) are "produced" in the fitting of activated fraction of particles (either activated fraction vs. supersaturation(SS) for particles of a given size or activated fraction vs. particle size at a given SS). The artifacts derived from DMA was calculated by Eq. 12 based on "volume-weighted diameter-specific perceived $\kappa_{app}$ values". I am not sure whether the artifacts in this study can reflect the real uncertainties in CCN measurement. Secondly, I am not sure whether the method used to calculate $\kappa_{app}$ and (and to derive $SS_{crit}$) is appropriate. Why the authors used "volume-weighted" approach? In my opinion, when the particle size distribution broadens, the number of both the larger particle and smaller particles increase in a largely similar rate. Then the ratio of activated particles (larger particles) to total particles (measured by CPC) as well as SScrit and kappa should be relatively invariant. Could the authors assess the uncertainties in kappa using the way that kappa is derived in the real CCN measurement?
3. In the CCN activation measurement, the supersaturation of CCN counter is often calibrated using the theoretical data of $(NH_4)_2SO_4$ or NaCl in the literature (Rose, Gunthe et al. 2008). The kappa of the standards ($(NH_4)_2SO_4$ or NaCl) and the sample aerosol would have the bias of the same direction. This may largely compensate the artifact of CCN measurement and thus lessen the role of instrument artifacts in the discrepancy between different measurements. It may be helpful to discuss this aspect.

Specific comments

1. L62, why do the authors particularly mention sea spray aerosol among various aerosol types?

2. L457, it is worth noting that these values are for the artifacts of CPC or CCN alone. The artifacts from CPC and CCN counting at high aerosol concentration counteract. Therefore, the combined effect of the CPC and CCN is much lower as the authors mentioned in L445-447.

   Please also state that these values ("$-0.57 < \kappa_{app,rtifact} < 0.42$") are for NaCl.

3. L137, the literature of the kappa values is not provided.
4. L202, L is not defined.
5. L216, the detailed motivation of design these 7 cases are not available (although they are mentioned in the conclusion section).
6. L253, it is not clear how exactly the $\kappa_{app, theory}$ (and $\kappa_i$) is derived. $\varepsilon_i$ and $\kappa_i$ are not defined. Please elaborate. And why do the authors use volume-weighted kappa?
7. L266-268, the artifacts due to the ratio of excess flow to sheath flow are not really discussed here, even less than in the abstract.
8. L412-416, why would "a distribution with a narrower peak than the one generated for this analysis be at risk for larger $\kappa_{app}$ artifacts for any total aerosol concentration…"?

Technical comments

1. I suggest numbering the section from the "Introduction".

**References**

Rose, D., S. S. Gunthe, et al. (2008). "Calibration and measurement uncertainties of a continuous-flow cloud condensation nuclei counter (DMT-CCNC): CCN activation of ammonium sulfate and sodium chloride aerosol particles in theory and experiment." Atmospheric Chemistry and Physics **8**(5): 1153-1179.

---

## Author Comment (AC1) · 9 Aug 2018

Author's response has been attached (please see Supplement).

Please also note the supplement to this comment:
https://www.atmos-meas-tech-discuss.net/amt-2018-164/amt-2018-164-AC1-supplement.pdf

---

## Referee Comment (RC2) · Anonymous Referee #2 · 16 Aug 2018

Review of the paper  amt-2018-164

Instrument Artifacts Lead to Uncertainties in Parameterizations of Cloud Condensation Nucleation
by
Jessica A. Mirrielees and Sarah D. Brooks

Possible artifacts in different types measurements of cloud condensation nuclei need to be discussed much more than has been done previously. This paper could thus be of large value for the scientific community. It is however important to make sure that the discussion is done in a way that is relevant to how CCN measurements normally are made, considering e.g. particle concentrations and data evaluation procedures. I recommend publication after major revision and the authors carefully considering the comments bellow. Since I think this paper needs major revision, I will not comment on small details, but focus on major issues and things that are repeated.

Even though more discussion about artifacts and good practice in CCN measurements are needed, this is not the first paper dealing with the issue and with the role of DMA flow ratio in determining the precision in fitting step functions for evaluating number of CCN as a function of supersaturation. I would recommend that this literature is summarized and that results in this paper are discussed in perspective of this literature.

Also, the use of hygroscopicity parameterizations goes much further back in time than indicated in this manuscript. The parameter epsilon is originally adopted from the work by Fitzgerald (1975) and in 1982 Fitzgerald et al. suggested a single hygroscopicity parameter (Bc).  There might be even older literature.

I also recommend that the authors state the limitations of the paper clearly: 1) All CCN measurements are not done with the set up indicated in figure 1. Lab studies can be performed with an SMPS instead of CPC if evaporation from the particles is suspected. 2) Field studies at low particle concentrations are made without a DMA upstream the CCNC. These measurements have there own issues.

Another limitation is that the paper does not treat all the important artifacts. An example is uncertainty in sizing due to evaporation of particle material or residual water in the particles while sized in the DMA. I can understand if this is out of scope for this article, but they could be mentioned. Other sources of errors are closer to the focus of this paper and could be included or at least references made to papers discussing them. I am mainly thinking about three effects: 1) the role of doubly charged particles, especially in lab studies in which atomized aerosols can be overcharged in comparison to equilibrium charge distribution and radioactive sources normally used are not strong enough to neutralize the aerosol. 2) Voltage offset in the DMA is sometimes an issue, especially when working with high supersaturations and small aerosol particle sizes. 3) The role of counting variability due to sampling statistics at low concentrations and how it influences the determination of SSc in different cases (for example different flow ratios).

I also have a comment that might sound nerdy, but I find it important that we stick to the definition of an aerosol as a population of solid and/or liquid particles and the

surrounding gas. Thus, we should not talk about aerosol size, when referring to the size of the particles. Please check the manuscript in line with this.

It is not clear how the critical supersaturation is determined: is it defined as the supersaturation at which #CCN/#CN is 0.5 or when its value is 50% of the level reached at high supersaturation (the later often being used in experimental work)? This will in some of the examples make a large difference, and need to be discussed. An example is line 346-348.

Please check the plots with particle size distributions. The y-axis should be $dN/dDp$ (with the unit $cm^{-3}$ $nm^{-1}$) if a linear diameter scale is used. Why are you using a linear diameter scale and not a logarithmic? Also, see my comments to figure 6 and 7 below. I think that these are critical for the quality of the paper and the conclusions!

Also, make sure that the figure captions and legends are sufficient.

Figure 6 b. As I understand it, the CPC counting limitations relevant here relate to the number concentration after the DMA. Are the DMA transfer function and the charge distribution taken into account when determining these curves? And if so, for which aerosol to sheath flow ratio are they made? Is it just a coincidence that the size distribution is cut at the same value of $dN/dDp$ (in $cm^{-3}$ $nm^{-1}$ ?) as the CPC concentration saturates (in the unit $cm^{-3}$). An how can the "saturated size distribution" be a horizontal line? Both charging probability and transfer function width (in a linear scale) are size dependent.

Figure 7 and the calculations behind them: How is $dN/dDp$ transferred into a concentration after the DMA? Which flows are used?

You use both saturation ratio and supersaturation in the theoretical discussion. As I can see you are using them correctly, but sometimes you use only saturation for saturation ration. I would recommend that you stick to saturation ratio in order to avoid confusing the readers.

Line 184. With a truly monodispers aerosol the concentration would also be 0.

The discussion and the conclusion section is mainly a repetition of the results (which might well be a part of these sections), but I would have liked to see a discussion on what should be considered good practice in CCN measurements, based on this work and the literature.

-Fitzgerald, J.W (1975) Approximation Formulas for the Equilibrium Size of an Aerosol Particle as a Function of Its Dry Size and Composition and the Ambient Relative Humidity J. Appl. Meteorol. 14, 1044-1049

-Fitzgerald, J.W.,Hoppel, W.A., and Vietti, M.A. (1982) The Size and Scattering Coeffi- cient of Urban Aerosol Particles at Washington, DC as a function of Relative Humidity J. Atmos. Sci. 39 1838-1852

---

## Author Comment (AC2) · 14 Sep 2018

Authors' response to comments on "Instrument Artifacts Lead to Uncertainties in Parameterizations of Cloud Condensation Nucleation" Referee #2

Authors' response: We thank the Reviewer for her/his feedback, which we feel have improved the manuscript. New sections have been added to the manuscript in order to address the recommended revisions. While it is beyond the scope of this manuscript to address very possible source of uncertainty in detail, we have added significantly to the discussion of errors. In particular, to address the Reviewer's concern that we had not included uncertainties which arise due to multiple charging in the DMA, we have now performed calculations and added a new figure and addition text to the manuscript. Multiple charging uncertainties are incorporated into our overall analysis and conclusions. Additional specific modifications are discussed below.

1) Reviewer comment: Even though more discussion about artifacts and good practice in CCN measurements are needed, this is not the first paper dealing with the issue and with **the role of DMA flow ratio in determining the precision in fitting step functions** for evaluating number of CCN as a function of supersaturation. I would recommend that this literature is summarized and that results in this paper are discussed in perspective of this literature.

Authors' response: We thank the reviewer for this suggestion. The previous literature regarding the role of flow ratio in precision of fitting the step functions include the works of Petters et al. (2007b), Rose et al. (2008) and Zhao-Ze and Liang (2014) which are now discussed in the text. To summarize, the DMA sample/sheath flow ratio increases, the DMA transfer function broadens, allowing particles that area both larger and smaller than the selected diameter to exit the DMA. This in turn broadens the CCN activated fraction curve (Rose et al., 2008). DMA transfer function broadening may be significant enough to also increase the concentration of multiply-charged particles downstream of the DMA, resulting in the appearance of "plateaus" in the activated fraction curves. An example of a plateau is shown in the figure below, where the green lines correspond with raw activated fractions and the dashed lines on each plot correspond with the activated fraction plateau for 25, 50, 100, and 200 nm diameter particles. This figure is further discussed in a new section in the manuscript, 3.1.3 Effect of double and triple charges on particles. The effects of multiple charges are addressed in more detail in the response to Question 4. A new section has also been added to the manuscript to address particles with multiple charges, 3.1.3 Effect of double and triple charges on particles.

[Figure]

Authors' changes to the manuscript: Page 21 Lines 379-390 the text now reads: "The aerosol/sheath ratio within the DMA also modulates the effect of multiple charges on $\kappa_{app}$. As the aerosol/sheath ratio increases, the transfer function broadens, allowing particles that area both larger and smaller than the selected diameter to exit the DMA. This in turn broadens the CCN activated fraction curve (Rose et al., 2008). The larger particles will activate as CCN at lower supersaturations than particles with the selected diameter, resulting in an increase in the activated fraction plateau due to multiple-charged particles and a further decrease in the determined $SS_{crit}$. Petters et al. 2007b showed that CCN activated fraction curves are significantly skewed by multiply-charged particles when the mode diameter of the aerosol population upstream of the DMA exceeds the critical diameter of the size-selected particles. In an example CCN activated fraction curve, Rose et al. 2008 demonstrated that a 1:6 ratio of doubly-to-singly charged particles resulted in an underestimated of the critical activation diameter by 2%. Zhao-Ze and Liang, 2014 also showed that multiply-charged particles can introduce significant uncertainty in values of kappa (~17%) into hygroscopicity calculations."

2) Reviewer comment: Also, the use of hygroscopicity parameterizations goes much further back in time than indicated in this manuscript. The parameter epsilon is originally adopted from the work by Fitzgerald (1975) and in 1982 Fitzgerald et al. suggested a single hygroscopicity parameter (Bc). There might be even older literature.

Authors' response: This is a good point, thank you. The Reviewer is referring to hygroscopicity parameterizations which precede the "kappa" notation but are nonetheless about hygroscopicity, and yes, we agreed that these references should be included. A number of references to earlier works have been added to the manuscript's introduction. Specific changes are described in the revised text below.

Authors' Changes to the manuscript:
Pages 4-5 Lines 49-66 the text now reads: "In earlier works on the hygroscopic growth of aerosol particles, particle composition was found to influence the relationship between particle growth and relative humidity. Covert et al., 1972 posited that the effect of relative humidity on the light-scattering properties of an aerosol could be used to determine its chemical composition, in conjunction with other chemical analysis techniques; this concept was employed in Charlson et al., 1974 to quantify the sulfate content in aerosol particles. Winkler 1973 developed an equation for approximating the growth of an aerosol particle with relative humidity, based on the quantity and physical characteristics of the soluble species in the particle. Another approximation for the relationship between the equilibrium size of a particle and relative humidity was derived by Fitzgerald in 1975, in which the soluble fraction and composition of the soluble component(s) are taken into account.
Parameterizations which use hygroscopicity to predict CCN activation that pre-date Petters and Kreidenweiss 2007 exist as well. Fitzgerald et al., 1982 derived a particle composition parameter using the mass fraction and physical properties the of soluble material in a particle. Kreidenweis et al., 2005 determined that the critical activation diameter of dry aerosol particles can be calculated from simplified Köhler theory using the physical properties of water and the solute in a solution droplet. This parameterization has been used in CCN closure studies (Bougiatioti et al., 2009; Moore et al., 2011; Moore et al., 2012). The earliest prediction of CCN concentrations for specific particle diameters and hygroscopicity used this parameterization as well (Mochida et al., 2006)."

3) Reviewer comment: I also recommend that the authors state the limitations of the paper clearly:

A) All CCN measurements are not done with the set up indicated in figure 1. Lab studies can be performed with an SMPS instead of CPC if evaporation from the particles is suspected.

B.) Field studies at low particle concentrations are made without a DMA upstream the CCNC. These measurements have their own issues.

A. Authors' response: We now state the scope of limitation of the paper clearly. The scope of this manuscript includes only CCN measurements that are used to calculate apparent hygroscopicity from critical supersaturation, which in turn is determined from activated fraction curves for monodisperse (or near-monodisperse) aerosol. Other setups can be used for CCN measurements, as the reviewer correctly points out. For example, if evaporation is a concern, an SMPS can be used to determine the size distribution of the aerosol particles. An SMPS can also be used in conjunction with unsized CCN measurements. A clarification has been added to section 3. Artifacts derived from CCN measurements (now titled "Artifacts derived from size CCN measurements").

Authors' changes to the manuscript:
Page 11 lines 173-178 the text now reads (additions are bolded): "CCN measurements **used for calculating apparent hygroscopicity from monodisperse aerosol** require accurate operation of three instruments: the CCN, the differential mobility analyzer (DMA), and the condensational particle counter (CPC). The scope of this manuscript includes only CCN measurements that are used to calculate apparent hygroscopicity from critical supersaturation, which in turn is determined from activated fraction curves for monodisperse (or near-monodisperse) aerosol. Other setups can be used for CCN measurements, as the reviewer correctly points out. For example, if evaporation is a concern, an SMPS can be used to determine the size distribution of the aerosol particles. An SMPS can also be used in conjunction with unsized CCN measurements."

B. Authors' response: The Reviewer is correct that a large number of field studies have been conducted without a DMA to size-select the aerosol prior to CCN measurements, Although these are somewhat different than the sized CCN measurements which are the focus of this paper, it is a good idea to mention them here. Examples which we now cite in the text include Jennings et al., 1996; Hudson and Xie, 1998; Modini et al., 2015; Duan et al., 2017; Schmale et al., 2018; Leng et al., 2013. However, our manuscript focuses on sized CCN measurements which may be used for the determination of $\kappa_{app}$. This is now clearly stated on page 10.

Authors' Changes to the manuscript:
Page 11 Lines 174-181 the text now reads: "This study considers sized CCN measurements which may be used for the determination of $\kappa_{app}$. In contrast, CCN studies are often conducted on the full ambient aerosol population without sizing the aerosol (Jennings et al., 1996; Hudson and Xie, 1998; Modini et al., 2015; Duan et al., 2017; Schmale et al., 2018; Leng et al., 2013).
Instrument artifacts will first be assessed separately for the DMA, CPC, and CCN counter. In the concluding section of the paper (and Fig. 10), the overall uncertainty due to the combination of these is presented and discussed."

4). Reviewer comment: Another limitation is that the paper does not treat all the important artifacts.

A. An example is uncertainty in sizing due to evaporation of particle material or residual water in the particles while sized in the DMA. I can understand if this is out of scope for this article, but they could be mentioned.

B. Other sources of errors are closer to the focus of this paper and could be included or at least references made to papers discussing them. I am mainly thinking about three effects. The first is the role of doubly charged particles, especially in lab studies in which atomized aerosols can be overcharged in comparison to equilibrium charge distribution and radioactive sources normally used are not strong enough to neutralize the aerosol.

C. Voltage offset in the DMA is sometimes an issue, especially when working with high supersaturations and small aerosol particle sizes.

D. The role of counting variability due to sampling statistics at low concentrations and how it influences the determination of SSc in different cases (for example different flow ratios).

A. Authors' response: While we cannot address all possible artifacts in this manuscript, a new section has been added to the manuscript to address sources of artifacts that are beyond the scope of the study, but should still be considered (3.1.4 Additional artifacts resulting from DMA measurements). We have included a short discussion of particle evaporation and growth due to volatility and hygroscopicity, respectively.

Authors' Changes to the manuscript:
Pages 21-22 Lines 394-402 the text now reads: "Several other factors that may impact experimental $\kappa_{app}$ are beyond the scope of this study, but are worth mentioning as they represent additional potential sources of error in some cases. First, volatile aerosols may partially evaporate inside the DMA, resulting in a decrease in particle size exiting the DMA. DMA sizing error due to aerosol volatility (defined as the ratio of sampled diameter to the selected diameter) increases with volatility, though sizing error can be decreased by increasing the sheath flow rate in the DMA. Conversely, hygroscopic aerosols may grow inside the DMA, resulting in larger particles existing the DMA. Operationally, errors in DMA sizing due to hygroscopic growth can be mitigated if aerosols entering the DMA inlet are in wet metastable states (higher aerosol RH at DMA inlet), and if DMA sheath flow rates are kept low (Khlystov, 2014)."

B. Authors' response: The reviewer has a valid point. On further consideration, we now include full consideration of one additional significant artifact, effects of multiple charging, in our analysis. We have added an analysis of multiple particle charges to the manuscript in a new section (3.1.3 Effect of double and triple charges on particles). In addition, 3 new figures have also been added (Figure S2 in supplement, Figures 5 and 6 in manuscript, all shown below).

Authors' Changes to the manuscript:
Pages 18-21 Lines 319-390 the text now reads: "The Grimm DMA employs a bipolar charger (also known as a neutralizer) to charge aerosol particles through the capture of gaseous ions. The analysis in Section 3.1.2 assumes that each particle carries a single (+1) charge. In reality, the methods used to charge particles prior to entering a DMA may impart two, three, or more charges to individual particles (**Fuchs, 1963**). The charge distribution resulting from a bipolar charger is roughly approximated using the Boltzmann law (**Keefe et al., 1959**). Additionally, the Boltzmann law assumes symmetric aerosol particle charging (equal concentrations of negatively and positively charged particles). Deviation from symmetric charging is observed in regions of high ionizations, and this deviation becomes more pronounced as particle size increases (**Hoppel and Frick, 1990**).

[revised manuscript text omitted]

**Figure S2** (a) Stationary charge distribution on particles shown in Fig. 3a. The particle diameters were chosen to represent particles that would be present due to single, double, and triple charging for the DMA selected diameters 25, 50, 100, and 200 nm. (b) Ratio of multiple charged particles to single charged particles.

[Figure]

**Figure 5** (a) Theoretical raw and adjusted activated fraction curves for single charged 25 nm particles, double charged 50 nm particles, and triple charged 75 nm particles; (b) single charged 50 nm particles, double charged 100 nm particles, and triple charged 150 nm particles; (c) single charged 100 nm particles, double charged 200 nm particles, and triple charged 300 nm particles; (d) single charged 200 nm particles, double charged 400 nm particles, and triple charged 600 nm particles. All particles are pure sodium chloride.

[Figure]

**Figure 6** (a) The critical supersaturation of sodium chloride particles determined from the activated fraction curves in Fig. 5. A sodium chloride curve calculated using κ-Köhler theory is shown for comparison. (b) $\kappa_{app}$ was calculated for the charge distribution; the gray dashed line indicates the literature value for the $\kappa_{app}$ of sodium chloride. (c) $\kappa_{app}$ artifacts resulting from multiple particle charges.

C. Authors' response: A new section has been added to the manuscript (3.1.4 Additional artifacts resulting from DMA measurements) that includes a short discussion of voltage shifts that may result from the space-charge field observed due to separation of the charged aerosol particles.

Authors' Changes to the manuscript:
Page 22 Lines 404-412 the text now reads: "Voltage shifts within the DMA (differences between the selected voltage and the actual voltage inside the DMA) can lead to discrepancies between selected and sampled particle diameters.  Voltage shifts may result from a space-charge field generated by the motion of charges within the DMA. Particles charged by the neutralizer will either be attracted towards or repelled away from the inner column of the DMA, depending on whether they are positively or negatively charged.  This charge separation creates a space-charge field which shifts the actual voltage within the DMA from the selected voltage.  The impact of the space-charge field on the midpoint and spread of the DMA transfer function increases as particle mobility increases (as particle size decreases), and as particle concentration increases **(Alonso and Kousaka, 1996; Alonso et al., 2000; Alonso et al., 2001)**."

D. Authors' response: New text has been added to the manuscript in order to address this issue (6. Counting statistics in CCN and CPC measurements).

Authors' Changes to the manuscript:
Pages 29 Lines 542-544 the text now reads: "Sampling at very low particle concentrations ($< 200$ cm$^{-3}$ total particle concentration) can introduce additional error into CCN and CPC measurements.  This error can be mitigated by increasing scan times.  For example, Moore et al., 2010 averaged CCN and particle concentrations over 5-second intervals for monodisperse particle concentrations $< 10$ cm$^{-3}$, and increased averaging time to  20-second intervals when the monodisperse particle concentration reached $< 6$ cm$^{-3}$."

5) Reviewer comment: I also have a comment that might sound nerdy, but I find it important that we stick to the definition of an aerosol as a population of solid and/or liquid particles and the surrounding gas. Thus, we should not talk about aerosol size, when referring to the size of the particles. Please check the manuscript in line with this.

Authors' response: the reviewer is correct about the technical difference between aerosols and particles, but we respectfully disagree that the term needs to be changed in the manuscript.  "Aerosol" is often used synonymously with "aerosol particle". In keeping with convention, we will not change the current wording. Doing so would likely make the text less clear.

6) Reviewer comment: It is not clear how the critical supersaturation is determined: is it defined as the supersaturation at which #CCN/#CN is 0.5 or when its value is 50% of the level reached at high supersaturation (the later often being used in experimental work)? This will in some of the examples make a large difference, and need to be discussed. An example is line 346-348.

Authors' response: Activated fraction data is fit with a sigmoid error function to determine the supersaturation at which 50 % of the particles have activated as CCN (activated fraction = 0.50), which is considered the operationally defined critical supersaturation $SS_{crit}$ (Rose et al., 2008). An alternative approach is to define critical supersaturation as the value of supersaturation corresponding to 50% of the maximum observed value.  However, this alternative method has a caveat in cases of external aerosol mixtures which include non-hygroscopic members.  In such cases, the highest CCN/CN reached will be much lower than 1 (due to the inactive member aerosols) and thus the resulting value of critical supersaturation would be lower that the SS required to the active aerosol members to become CCN.  A clarification has been added to the Background section of the manuscript.

Authors' changes to the manuscript: Page 8 Lines 125-127 the text now reads (clarification bolded):
"Activated fraction data is fit with a sigmoid error function to determine the supersaturation at which

50 % of the particles have activated as CCN (**activated fraction = 0.50**), which is considered the operationally defined critical supersaturation $SS_{crit}$ (Rose et al., 2008)."

Additional clarification was added to Section 4.1 CPC operation at low concentration.

Page 21 Lines 410-411 the text now reads: "Critical supersaturation was determined for each CPC case by finding the percent supersaturation at which activated fraction = 0.50."

7) Reviewer comment: Please check the plots with particle size distributions. The y-axis should be dN/dDp (with the unit cm-3 nm-1) if a linear diameter scale is used. Why are you using a linear diameter scale and not a logarithmic? Also, see my comments to figure 6 and 7 below. I think that these are critical for the quality of the paper and the conclusions!

Authors' response: We see the Reviewer's point here. Employing dN/dDp is one of the standard way to plot an aerosol size distribution and readers are likely to be comfortable viewing such graphs. It is advantageous to view instrumental data in dN/dlogDo format, in which the normalization accounts for differences in the widths of size bins employed by various instruments and in various size ranges. However, our analysis is based on a theoretical aerosol distribution in which there are no disparate ranges in size bins to account for, and thus normalization is not necessary. For the purposes of this analysis, it is more straightforward to plot concentration in lieu of dN/dlogDp.

In the original text, logarithmic scales were used for particle size distribution plots in Figures 8a and 8b (previously Figures 6a and 6b) and in Figure 9b (previously Figure 7b). A linear scale is used in Figures 3a and 3c because we feel that it clearly demonstrates "visual multiplication" (combining the distribution in 3a with the transfer function for 100 nm particles in 3b) performed in our analysis. In addition, given the small range of concentrations observed in Figure 3c, this is the clearest presentation of the data.

8) Reviewer comment: Also, make sure that the figure captions and legends are sufficient.

Authors' response: Several figure captions have been edited for clarity as specified below.

Authors' changes to the manuscript: figure captions have been changed as indicated below (changes are shown in bold).

"**Figure 1** Experimental setup **used for obtaining sized** CCN and particle concentration measurements **from an aerosol sample.**"
"**Figure 2** Simplified flow diagram of a DMA with an inner electrode radius $r_1$, outer electrode radius $r_2$, distance between aerosol inlet and sample outlet $L$, **clean sheath air flow $Q_{sh}$, aerosol flow $Q_a$, excess air flow $Q_e$, and sample air flow $Q_s$.**"
"**Figure 3** (a) A theoretical aerosol distribution generated using a lognormal function centered at 50 nm. (b) The transfer function calculated using Eq. (7). (c) **Multiplying the distribution by the transfer function gives the downstream aerosol concentration (cm$^{-3}$).**"
"**Figure 4** (a) Apparent hygroscopicity $\kappa_{app}$ for **DMA cases 1-7 for sodium chloride (triangles) and ammonium sulfate (circles) [see legend in (b)].** (b) Critical supersaturation of ammonium sulfate and sodium chloride particles calculated using $\kappa_{app}$ values derived in (a). Ammonium sulfate and sodium chloride curves from κ-Köhler theory are shown for comparison. **Legend colors apply to both salts**. (c,d) DMA-flow-derived artifacts in $\kappa_{app}$ are shown for each DMA case and both salts."
"**Figure 7** (a) Counting efficiency curves for CPC **Cases 1-6 (shown in Table 3).**

(b-e) CCN activated fraction curves for 25, 50, 100, and 200 nm NaCl, respectively. (f) Critical supersaturation calculated for each **particle diameter**. (g) Theoretical $\kappa_{app}$ for each CPC case **and particle diameter.** (h) Artifacts in $\kappa_{app}$ for each CPC case **and particle diameter.**"

"**Figure 8** (a) Theoretical relationships between the reference aerosol concentration and CPC concentration. (b) Concentration-dependent counting efficiencies **from (a)** were applied to four theoretical aerosol distributions. (c-f) Activated fraction curves for **CPC Distribution 1 and** particle diameters 25, 50, 100, and 200 nm NaCl aerosol, respectively. (g,h) Critical supersaturation and $\kappa_{app}$ for each case. (i) Artifacts in $\kappa_{app}$ for each case."

"**Figure 9** (a) Counting efficiencies of the DMT CCN-100 **for specific supersaturations**. (b) Lognormal aerosol distributions used to study CCN undercounting at high concentrations. (c-f) Activated fraction curves for 25, 50, 100, and 200 nm NaCl particles. Supersaturation-specific counting efficiencies from (a) applied to theoretical sigmoid curves for NaCl CCN activation. Activated fraction in the case of 100 % counting efficiency is shown for comparison. (g) Critical supersaturation for each case. (h) Theoretical $\kappa_{app}$ calculated for each case. (i) Artifacts in $\kappa_{app}$ artifacts for each case."

9) Reviewer comment: Figure 6 b. As I understand it, the CPC counting limitations relevant here relate to the number concentration after the DMA.

   A. Are the DMA transfer function and the charge distribution taken into account when determining these curves? And if so, for which aerosol to sheath flow ratio are they made?

   B. Is it just a coincidence that the size distribution is cut at the same value of dN/dDp (in cm-3 nm-1 ?) as the CPC concentration saturates (in the unit cm-3). And how can the "saturated size distribution" be a horizontal line? Both charging probability and transfer function width (in a linear scale) are size dependent.

   A. Authors' response: Yes, the CPC instrument is positioned after the DMA. The DMA transfer function is not taken into account in the CPC concentration analysis here.

   Page 11 Lines 179-181 the text now reads: "Instrument artifacts will be assessed separately for the DMA, CPC, and CCN counter.   In the concluding section of the paper (and Fig. 10), the overall uncertainty due to the combination of these is presented and discussed."

   B. Authors' response:  This is not a coincidence; this concentration is the concentration at which the CPC can no longer count additional particles (and is not related to charging probability or the transfer function).  An additional sentence has been added to the manuscript for clarification.

   Page 26 Lines 477-479 the text now reads: "It should be noted that the CPC concentration (the concentration that would be measured and recorded by the CPC in Cases 7-10) levels off at the saturation concentration for each case."

10) Reviewer comment: Figure 7 and the calculations behind them: How is dN/dDp transferred into a concentration after the DMA? Which flows are used?

    Author's response: To clarify, we have only used concentration in this manuscript, not dN/dDp.  If we understand this question correctly, the reviewer is referring to the fact that particles are lost in the DMA which leads to error in particle concentration measured by the CPC placed downstream of the DMA.  Indeed, particle losses within the DMA are substantial. If aerosol concentration were the

measurement objective then a more accurate measurement of concentration would simply use a CPC, thus minimizing particle losses. However, for the purposes of this manuscript, we consider activated fraction (the ratio of particles measured by the CCN instrument to those measured by the CPC for a given particle diameter and supersaturation). In this case, losses in the DMA impact the CPC and CCN concentrations exactly the same way, so no error in introduced in the resulting activated fraction. In this analysis, the actual CCN and aerosol particle concentrations are only used for determining how activated fraction is effected in each CPC or CCN case, if at all. We acknowledge that estimation of particle losses in the DMA, and between the DMA and CPC/CCN, are worth considering but the treatment of these particle losses is beyond the scope of this manuscript.

11) Reviewer comment: You use both saturation ratio and supersaturation in the theoretical discussion. As I can see you are using them correctly, but sometimes you use only saturation for saturation ration. I would recommend that you stick to saturation ratio in order to avoid confusing the readers.

11. Authors' response: We thank the reviewer for finding this error. All mentions of "saturation" have been revised to "saturation ratio".

12) Reviewer comment: Line 184. With a truly monodisperse aerosol the concentration would also be 0.

12. Authors' response: This is true, and the sentence indicated has been changed for clarification.

Page 13 Lines 212-213 the text now reads: "In other words, $Q_s$ would ideally consist only of aerosols with diameters equal to, or very nearly equal to, the selected diameter."

13) Reviewer comment: The discussion and the conclusion section is mainly a repetition of the results (which might well be a part of these sections), but I would have liked to see a discussion on what should be considered good practice in CCN measurements, based on this work and the literature.

Authors' response: Good practice in CCN measurements was originally addressed in the Conclusion section of the original manuscript, in terms of minimizing kappa artifacts. Based on the reviewer's comment, we have expanded on the discussion and conclusions, including references to DMA sample/sheath ratios recommended in other studies for comparison.

Authors' changes to manuscript:

Page 32 Lines 580-584 the text now reads: "These demonstrate that limiting $Q_a/Q_{sh}$ to $\leq 0.10$ will result in a narrow particle size distribution downstream of the DMA. Other studies have recommended employing DMA sample/sheath ratios of 0.2 (Petters et al., 2007; Carrico et al., 2008; Moore et al., 2010) or 0.1 (Moore et al., 2010; Zhao-Ze and Liang, 2014) in order to minimize measurement aerosols due to transfer function broadening."

Page 32 Lines 586-591 the text now reads: "The effects of multiply-charged particles on $\kappa_{app}$ calculations were also quantified. Small, positive $\kappa_{app}$ artifacts ($1 - 3$ % of $\kappa_{app}^{NaCl}$) were observed when particles with +2 and +3 charges were not accounted for. This analysis considered a theoretical aerosol distribution in which most of the particles measure less than 100 nm in diameter. Actual aerosol distributions vary temporally and spatially, and often include accumulation and coarse modes that would result in larger $\kappa_{app}$ artifacts."

Page 33 Lines 611-615 the text now reads (additions are bolded): The combined artifacts for the cases where the highest artifacts were observed (DMA Case 4, **multiple particle charging**, CPC Case 4, CPC Case 10, CCN Case 1) are 0.21, 0.24, 0.32, and 0.21 for 25, 50, 100, and 200 nm particles respectively, as shown in Fig. 10. The combined artifacts for the lowest-artifact cases (DMA Case 2, CPC Case 3, and CCN Case 4) are $< 0.001$ except for 200 nm particles, where $\kappa_{app,artifact} = 0.0013$.

Page 36 Lines 670-672 the text now reads (**additions are bolded**): "Under optimal operating conditions, where the DMA sample/sheath ratio is 0.10 and excess/sheath ratio is 1.0, and in the absence of undercounting by the CPC or CCN, uncertainties in $\kappa_{app}$ are within ±1.1 % for 25 to 200 nm particles. When the DMA sample/sheath ratio drops to 0.05, $\kappa_{app}$ uncertainties decrease to ±0.01 %. Additionally, errors in activated fraction (and therefore $\kappa_{app}$) resulting from the bipolar charge distribution can be corrected by determining the fraction of particles with multiple charges.

---

## Author Response (AR1)

Authors' response to referee comments on "Instrument Artifacts Lead to Uncertainties in Parameterizations of Cloud Condensation Nucleation"

Referee #1

We thank the Reviewer for her/his detailed suggestions which we feel have improved the manuscript. Specific modifications are discussed below.

**General Comments**

 Reviewer comment: This study used a pure theoretical approach to assess the artifacts in various CCN operating scenarios. However, many scenarios are not common in the real CCN activation measurement.

A. For example, is very rare that the particle number concentrations at the output of DMA reach 1e4 # cm-3, or even 5e6 # cm-3 as investigated in the section of artifacts derived from CPC and CCN.

B. The authors suggested in the introduction section that the discrepancy in experimental results for ammonium nitrate and some organics in the literature are contributed by the artifacts in CCN measurement. An interesting question is to what extent the artifacts investigated here can explain the discrepancies in the kappa of ammonium nitrate, for example, in the literature.

A. Authors' response: As the Reviewer correctly points out, the scenarios in this study include conditions which are rare, as well as more typical conditions. Specifically, a DMA output aerosol concentration of  $\sim 5.6 \times 10^4 \ cm^{-3}$  is very high. We intentionally include this concentration, because such conditions have been encountered in the field under certain conditions, such as new particle formation events (Hameri, O'Dowd, and Hoell 2002). Also, in our original manuscript, we attempted to clearly qualify all conclusions according to "standard" vs. "high concentration" situations. We have now modified the text to make these delineations clearer, as specified below. In order to assess CPC operation at more typical concentrations, as well as the initially assessed  $5 \times 10^6 \ cm^{-3}$  total particles, three more total particle distributions have been added to the analysis in Section 4.2 CPC operation at high concentration.

Authors' changes in manuscript:

We have added the following statements for clarification:

Page 24-25 Lines 449-452 the text now reads: "CPC undercounting issues which arise even at relatively low concentrations (which one would expect to encounter under standard experimental conditions) will be discussed in this section. Concentration-dependent effects encountered at higher concentrations will be explored in Sect. 4.2."

Pages 26-27 Lines 488-496 the text now reads: "In order to assess the importance of undercounting in CPC Cases 7-10, four theoretical aerosol distributions with a peak concentration at 50 nm were employed (Table 5, Fig. 8b). CPC Distribution 1 represents a worst-case scenario of similar magnitude to the highest particle concentrations measured during a coastal nucleation event (Hameri et al., 2002; Sem, 2002), while CPC Distributions 2, 3, and 4 are lower in concentration (due to the lack of undercounting in CPC Distributions 2, 3, and 4 as demonstrated in Figure 6b, the remaining analysis for CPC operation at high concentration considers only CPC Distribution 1.) CPC Cases 8-10 were applied to CPC Distribution 1 in order to determine the concentration measured by the CPC

for 25, 50, 100 and 200 nm aerosols. The counting efficiency was then calculated for each case and aerosol diameter in CPC Distribution 1."

Page 29 Lines 541-543 the text now reads: "Note that CCN Cases 1-4 are identical to the aerosol distributions. CPC Distributions 1-4 used for the high-concentration CPC cases."

Page 33 Lines 605-608 the bolded section was added to the following sentence: "In contrast,  $\kappa_{app}$  artifacts are negligible (< 0.10 % of  $\kappa_{app}^{NaCl}$ ) in CPC Case 3, where maximum counting efficiency = 100 %. CPC Cases 8 and 10 (**applied to the highest-concentration case, CPC Distribution 1**) represent undercounting at high concentration with CPCs where saturation is observed at 4 × 104 cm-3 and 1 × 104 cm-3, respectively."

Page 33 Lines 611-613 the text now reads: "It should be noted that undercounting was only observed for one of the four upstream distributions studied, CPC Distribution 1. No undercounting was observed when CPC Cases 7-10 were applied to CPC Distributions 2-4."

Page 55: CPC distributions for operation at high concentration have been added to Table 5, which in turn has been moved earlier in the manuscript to coincide with the first mention of these aerosol distributions. These CPC distributions are identical to the distributions used for CCN Cases 1-4.

Page 56: Three more (total aerosol) distributions have been added to Figure 8b in order to demonstrate the effect of CPC undercounting on lower aerosol concentrations. Figures 8c-i are only applied to the first distribution, CPC Distribution 1, because this is the only distribution in which undercounting occurs.

B. Authors' response: The ammonium nitrate kappa value comes from Svenningsson et al 2006 (Hygroscopic growth and critical supersaturations for mixed aerosol particles of inorganic and organic compounds of atmospheric relevance), in which the DMA sample/sheath ratio was maintained between 1.2 and 2, and the DMA-selected diameters fell in the range 50-180 nm. The apparent hygroscopicity for ammonium nitrate was found to be 0.577-0.753, with a mean value of 0.67. If 0.67 is assumed to be the true kappa for ammonium nitrate, then this sample/sheath ratio could lead an experimental kappa as low as 0.65 or as high as 0.69, which would not fully explain the actual experimental range. This assessment ignores possibility of under/over counting which could introduce additional errors. Since CPC and CCN spectrometer concentrations are not discussed in Svenningsson et al 2006, we cannot evaluate the likelihood of under/overcounting. Alternatively, the observed range of values could also arise (to some degree, at least) from fitting the activated fraction data in order to calculate kappa.

2) Reviewer comment: The approach used to derive artifacts from DMA in this study is significantly different from the real CCN measurement.

A. Firstly, in the real CCN measurement, uncertainties in SScrit (accordingly kappa) are "produced" in the fitting of activated fraction of particles (either activated fraction vs. supersaturation(SS) for particles of a given size or activated fraction vs. particle size at a given SS). The artifacts derived from DMA was calculated by Eq. 12 based on "volume-weighted diameter-specific perceived  $\kappa app$  values". I am not sure whether the artifacts in this study can reflect the real uncertainties in CCN measurement. Could the authors assess the uncertainties in kappa using the way that kappa is derived in the real CCN measurement?

B. Secondly, I am not sure whether the method used to calculate kapp and (and to derive SScrit) is appropriate. Why the authors used "volume-weighted" approach? In my opinion, when the particle size distribution broadens, the number of both the larger particle and smaller particles increase in a largely similar rate. Then the ratio of activated particles (larger particles) to total particles (measured by CPC) as well as SScrit and kappa should be relatively invariant.

A. Authors' response: To clarify, the uncertainties in critical saturation that arise from fitting the activated fraction data may be produced by several physical factors. First, the size of the aerosols will affect the fit. A broader size distribution will lead to a larger standard deviation in the sigmoid curve fit. Secondly, composition will also affect the standard deviation in the fit; a pure aerosol (one compound) may be fit with a sigmoid curve characterized by a smaller standard deviation, and the sigmoid curve fit for a mixture will have a larger standard deviation. Pure compounds were analyzed in this theoretical study, so the broadened size distribution was the only factor taken into account. The uncertainties in critical supersaturation are determined experimentally rather than theoretically in CCN experiments.

B. Authors' response: The volume-weighted approach accepted as a standard convention of kappa theory. The kappa for a mixture consisting of *i* components is calculating using a volume-weighted mixing rule (Petters and Kreidenweis 2007):

$$\kappa = \sum_{i} \epsilon_{i} \kappa_{i}$$

Where  $\epsilon_i$  is the volume fraction of each component, and  $\kappa_i$  is the hygroscopicity of each component. In this case, a pure compound is used instead of a mixture; instead, each component is an aerosol with a specific diameter, and each  $\kappa_i$  is the perceived kappa that would be calculated using the diameter selected at the DMA.

When the particle distribution broadens, smaller and larger electrical mobilities are favored equally/symmetrically, but this is not the case for smaller and larger particles. We have included Figure 3 from my paper below as an illustration. Figure 3b demonstrates that the transfer function is broadened more dramatically for larger diameters than for smaller diameters, and that this effect increases as the aerosol/sheath flow increases in the DMA (aerosol/sheath ratios: blue line = 0.05, black line = 0.1, mint green line = 0.2, yellow-green line = 0.3). Under recommended operating conditions (aerosol/sheath ratio  $\leq 0.1$ ), this effect is relatively small.

The aerosol distribution at the DMA inlet must be considered as well. A theoretical distribution was used in this analysis (Fig. 3a below). In this distribution, aerosols larger than the selected diameter would be favored over smaller aerosols if the selected diameter is less than 50 nm (the distribution peak); if the selected diameter is larger than 50 nm, smaller aerosols would be overrepresented compared to larger aerosols. This effect would be diminished if the aerosol distribution at the DMA sample inlet was broader than the theoretical distribution shown; conversely, this effect would be more dramatic if the aerosol distribution was narrower than the one shown.

3) Reviewer comment: In the CCN activation measurement, the supersaturation of CCN counter is often calibrated using the theoretical data of (NH4)2SO4 or NaCl in the literature (Rose, Gunthe et al. 2008). The kappa of the standards ((NH4)2SO4 or NaCl) and the sample aerosol would have the bias of the same direction. This may largely compensate the artifact of CCN measurement and thus lessen the role of instrument artifacts in the discrepancy between different measurements. It may be helpful to discuss this aspect.

Authors' response: This is an interesting point, and the bias in the kappa determined from the standards should be in the same direction as the bias in an experimental CCN measurement. However, the magnitude of this bias may not be the same for the standards and the aerosol studied in a CCN experiment (for pure compounds or mixtures). The biases ( $\kappa_{app,theory} - \kappa_{app,literature}$ ) observed in this study, as shown below in Figure 4, are not equal for the two ionic compounds studied. The relative bias may be similar (for example, if the experimental apparent hygroscopicity of sodium chloride is calculated to be 10% higher than the true value, one might expect ~10% overestimation of the apparent hygroscopicity of an aerosol sample calculated using data from the same CCN instrument). This would not necessarily lead to 100% compensation in the experimentally-determined apparent hygroscopicity for the aerosol sample being studied. However, some compensation would be reasonable to expect. This study considers two compounds that are used as standards for CCN instrument calibration. A further analysis could apply these results to an experimental setting, in which such compensation is discussed.

Authors' changes in manuscript:

Page 34 Lines 627-631 the text now reads: "We note that Fig. 4c-d demonstrated that  $\kappa_{app}$  error may result from instrument artifacts for ammonium sulfate and sodium chloride, two standard compositions used in calibration of CCN instruments (Rose et al., 2008). Therefore, the  $\kappa_{app}$  error encountered while calibrating the CCN instrument may compensate for the CCN measurement bias of aerosol samples. However, as also demonstrated in Fig. 4c-d, the magnitude of this instrumentally-derived bias varies by compound.

Specific comments

1. Reviewer comment: L62, why do the authors particularly mention sea spray aerosol among various aerosol types?

Authors' response: Sea spray is an important natural source of aerosols, but our initial discussion left out other common aerosol types. To broaden the discussion, a reference has been added to another study which summarizes the apparent hygroscopicity values of continental aerosols (Andreae and Rosenfeld 2008).

Authors' changes in manuscript:

Page 6, Lines 79-81, the text now reads: "Another study, which included a survey of observational CCN data, proposed that marine and continental aerosols could be described by  $\kappa_{app}$  values of 0.7  $\pm$  0.2 and 0.3  $\pm$  0.1 respectively (Andreae and Rosenfeld 2008)."

2. Reviewer comment: L457, it is worth noting that these values are for the artifacts of CPC or CCN alone. The artifacts from CPC and CCN counting at high aerosol concentration counteract. Therefore, the combined effect of the CPC and CCN is much lower as the authors mentioned in L445-447. Please also state that these values (" $-0.57

---

## Referee Report (RR1)

The authors have addressed most of my comments and the manuscript has improved. However, a few major concerns remain.

General comments

1.  The authors responded to my comment.
    For Part B, I suggest that this response regarding to what extent the artifacts investigated here can explain the discrepancies in the kappa in the literature should be also incorporated into the revised manuscript.

2.  Regarding my formed general comment 2:
    a)  I think that the assumption of the "volume-weighted approach accepted as a standard convention of kappa theory" is that different components are internally mixed on particles! Particles at different sizes are apparently not internally mixed. Taking an extreme example, if an aerosol population consists of black carbon particles for all particles <100 nm and $(NH_4)_2SO_4$ particles for all particle >100 nm. If one measure $D_{50}$ at 0.1% one would get a $D_{50}$ of ~140 nm since the $D_{50}$ of $(NH_4)_2SO_4$ is ~140 nm. And from that $D_{50}$ vs. SS, one would get a kappa of ~0.6 (the kappa value of $(NH_4)_2SO_4$). Of course, the $D_{50}$ vs. SS data sets obtained in this case will not fall on the lines of constant kappa.
    b)  This again leads to my concern about the approach used to derive artifacts from DMA in this study because it is different from the real CCN measurement. To clarify my comment, in the real CCN measurement, at first activated fractions of particles are obtained, either activated fraction vs. supersaturation(SS) for particles of a given size or activated fraction vs. particle size at a given SS. The activation curve is then fitted to derive a $D_{50}$ or $SS_{50}$ and from $D_{50}$ vs. SS or D vs. $SS_{50}$, kappa is obtained. In order to investigate the uncertainties of kappa due to instrument artifacts, one would need simulate the data acquisition process of CCN activation measurement by simulating the number of particles and number of activated particles in each size bin (in the case of $D_{50}$ vs. SS) and then activation fraction and $D_{50}$. I am not sure whether the artifacts in this study can reflect the real uncertainties in CCN measurement and is useful to get an idea of the uncertainties in CCN measurement.

3.  Regarding my formed general comment 3:
    I think one would like to see a more quantitative analysis of the uncertainties of kappa after taking the influence of instrument calibration by a standard compound into account. This is most relevant to real CCN measurement and is most interesting to those who do the measurement and who use these data.
    I also suggest that the discussion should be somewhat included in the conclusion because if after the calibration using $(NH_4)_2SO_4$ or NaCl, the discrepancy caused by instrument artifacts would be much smaller than values shown in the manuscript.

Technical comments

1.  In figure 3a, in the lognormal distribution it is dN/dlogDp that follows the shape of the curve rather than N (number concentration, y-axis) or dN/dDp.

---

## Author Response (AR2)

**Authors' response to comments on "Instrument Artifacts Lead to Uncertainties in Parameterizations of Cloud Condensation Nucleation", Revised Submission, Referee #1**

**Authors' response: We thank the Reviewer for her/his detailed feedback. In particular, we agree with the Reviewer that our assumptions regarding volume weighted components of kappa warranted revision and we have now revised that section accordingly. Specific modifications are discussed below.**

**General Comments**

1) **Reviewer comment:** The authors responded to my comment. For Part B, I suggest that this response regarding to what extent the artifacts investigated here can explain the discrepancies in the kappa in the literature should be also incorporated into the revised manuscript.

   **Authors' response:** Agree. This was an oversight on our part. We have now added this discussion to the manuscript.

   **Authors' changes to the manuscript:** Page 18 Lines 320-325 the text now reads: "This analysis was also applied to the range of apparent hygroscopicity values Svenningsson et al., 2006 found for ammonium nitrate $0.577 \leq \kappa_{app} \leq 0.753$, with a mean value of 0.670. If 0.670 is assumed to be the true $\kappa_{app}$ for ammonium nitrate, then the sample/sheath ratio used to determine $\kappa_{app}$ (1.2-2 L min$^{-1}$) could lead an experimental kappa as low as 0.665 or as high as 0.674, which would not fully explain the actual experimental range. This assessment ignores possibility of under/over counting which could introduce additional errors."

2) **Reviewer comment:** Regarding my formed general comment 2

   A. I think that the assumption of the "volume-weighted approach accepted as a standard convention of kappa theory" is that different components are internally mixed on particles! Particles at different sizes are apparently not internally mixed. Taking an extreme example, if an aerosol population consists of black carbon particles for all particles <100 nm and (NH4)2SO4 particles for all particle >100 nm. If one measure D50 at 0.1% one would get a D50 of ~140 nm since the D50 of (NH4)2SO4 is ~140 nm. And from that D50 vs. SS, one would get a kappa of ~0.6 (the kappa value of (NH4)2SO4). Of course, the D50 vs. SS data sets obtained in this Case will not fall on the lines of constant kappa.

   B. This again leads to my concern about the approach used to derive artifacts from DMA in this study because it is different from the real CCN measurement. To clarify my comment, in the real CCN measurement, at first activated fractions of particles are obtained, either activated fraction vs. supersaturation(SS) for particles of a given size or activated fraction vs. particle size at a given SS. The activation curve is then fitted to derive a D50 or SS50 and from D50 vs. SS or D vs. SS50, kappa is obtained. In order to investigate the uncertainties of kappa due to instrument artifacts, one would need simulate the data acquisition process of CCN activation measurement by simulating the number of particles and number of activated particles in each size bin (in the Case of D50 vs. SS) and then activation fraction and D50. I am not sure whether the artifacts in this study can reflect the real uncertainties in CCN measurement and is useful to get an idea of the uncertainties in CCN measurement.

**A. Authors' response:** The reviewer is correct, and we thank the reviewer for identifying this flaw in our previous analysis. Specifically, equation 12, read

$$\kappa_{app,theory} = \sum_i \epsilon_i \kappa_i \tag{12}$$

where $\epsilon_i$ is the volume fraction of aerosol of each diameter $i$, and $\kappa_i$ is the perceived $\kappa_{app}$ for each diameter (adapted from Petters and Kreidenweis [2007]). In this equation, there is an underlying assumption that all particles of a certain size have a certain composition, and that variations in composition (and thus kappa) occur only with corresponding variations in size. Since this assumption doesn't hold true for all aerosol populations, we have revised the analysis to include the active fraction as described below.

**B. Authors' response:** The conceptual changes detailed in 2A have been applied to the assessment of apparent hygroscopicity artifacts derived from DMA measurements.

**Authors' changes to the manuscript to address the issues raised in 2A&B:** In addition to the text below, Figures 4 and 10 have been updated, and Supplemental Figure S2 has been added.

Pages 16-17 Lines 272-317 the text now reads: "To test how uncertainties in DMA diameter translate to uncertainties in $\kappa_{app}$, the activation of particles downstream of the DMA was assessed. First, for each case and diameter (25, 50, 100, and 200 nm) the critical saturation ratio $s_{crit}$ was calculated for each particle diameter range downstream from the DMA using Eq. 3a. These critical saturation ratios were converted to critical percent supersaturation $SS_{crit}$ and used to calculate the activated fraction $AF$ for the aerosol particles downstream from the DMA for percent supersaturations $0.01 < SS < 1.5$, using the equation:

$$AF = \frac{1}{2}\left(1 + \text{erf}(\frac{SS - SS_{crit}}{\sigma\sqrt{2}})\right) \tag{12}$$

[revised manuscript text omitted]

Added Figure S2 Exemplary $(NH_4)_2SO_4$ CCN activation curves for DMA Cases 1-7.

3) **Reviewer comment:** Regarding my formed general comment 3: I think one would like to see a more quantitative analysis of the uncertainties of kappa after taking the influence of instrument calibration by a standard compound into account. This is most relevant to real CCN measurement and is most interesting to those who do the measurement and who use these data. I also suggest that the discussion should be somewhat included in the conclusion because if after the calibration using (NH4)2SO4 or NaCl, the discrepancy caused by instrument artifacts would be much smaller than values shown in the manuscript.

**Authors' response:** As the Reviewer points out, accuracy in measurements depend on accurate calibrations. In this case, accurate determination of the supersaturation setpoints within the CCN instrument are dependent on accurate sizing of aerosols entering the CCN, and therefore are dependent on the DMA sizing during CCN calibration. CCN calibrations during two standard compounds, ammonium sulfate and sodium chloride, as described in detail in Rose (2008). Fortunately, if the calibration procedure described by Rose is followed and an optimal DMA aerosol to sheath ratios employed, the uncertainties will be minimal. Specifically, our analysis shows that an aerosol to sheath ratio of 1:10 or 1:20 (Case 1 or 2, respectively) is recommended for all CCN calibrations. This will result in kappa uncertainties of less than 1% for all dry sizes (25 to 200 nm). However, if CCN calibrations are performed using a DMA operated with less than ideal aerosol to sheath ratios, substantial errors will be introduced. Analysis of the impact of DMA uncertainties on CCN calibrations are discussed in detail in the Supplemental Materials. In the worst case scenario amongst the cases evaluated here (Case 4), the resulting uncertainty in apparent kappa is 15%.

Details of the assessment of impacts of DMA sizing on CCN calibration follow:

We have now assessed the effects of calibration with a standard compound on subsequent CCN measurements, given that the DMA flow settings used in the calibration are the same as those used for subsequent measurements. Calibration with a standard will yield new parameters, $A$ and $B$, for a linear equation that relates percent supersaturation, $\%SS$, to the change in temperature set by the instrument, $\Delta T_{set}$:

$$\Delta T_{set} = (A \times \%SS) + B$$

If the slope $A$ and/or y-intercept $B$ are inaccurate, the instrument will choose $\Delta T_{set}$ incorrectly for input percent supersaturation values. In order to model this error, the $\kappa$-Köhler theory $\%SS_{crit}$ for 25, 50, 100, and 200 nm ammonium sulfate and sodium chloride was used to determine the "correct" $\Delta T_{set}$ for the CCNC, using parameters $A$ and $B$ from a previous CCNC calibration in our lab. These $\Delta T_{set}$ were then paired with the $\%SS_{crit}$ for DMA Cases 1-7 determined in Section 3.1.2 $\kappa_{app}$ artifacts arising from DMA flow ratios, as shown in Figure S3. A linear regression was run for each composition and DMA Case to find new parameters $A$ and $B$. For clarity, the original $\Delta T_{set}$ equation with the original parameters $A_0$ and $B_0$ will hereon be referred to as Eq. S1, and the new $\Delta T_{set}$ equations for each DMA Case $C$ with new parameters $A_C$ and $B_C$ will be referred to as Eq. S2.

$$\Delta T_{set} = (A_0 \times \%SS) + B_0 \qquad\qquad (S1)$$

$$\Delta T_{set} = (A_C \times \%SS) + B_C \qquad\qquad (S2)$$

For each composition (sodium chloride or ammonium sulfate) and DMA Case, Eq. S2 was used to determine the $\Delta T_{set}$ that the CCNC would set to achieve a series of percent supersaturations (0.01-1.5%). Then, Eq. S1 was used to determine the actual percent supersaturation that would result from each $\Delta T_{set}$. A few assumptions have been made so far: first, that the DMA aerosol/sheath ratio that was used during the calibration is also used in order to collect CCN activation data later, using the same compound; and second, that the original $A_0$ and $B_0$ used Eq. S1 were correct.

Two activated fraction curves were then plotted for each DMA Case (an example with DMA Cases 1-4 is shown in Figure S4). The activated fraction values for both curves were taken from the results in Section 3.1.2 $\kappa_{app}$ artifacts arising from DMA flow ratios. The accurate percent supersaturation values (dashed lines) were obtained from the original equation, and the observed percent supersaturation values (solid lines) are the values that would be reported by the CCNC according to the new equations.

Then, $\%SS_{crit}$ was determined for each observed activated fraction curve, as shown in Fig. S5a. The apparent hygroscopicity $\kappa_{app}$ was calculated for each DMA Case using Eq. 4, as shown in Fig. S5b. Apparent hygroscopicity artifacts are shown in Fig. S5c-d.

**Authors' changes to the manuscript:** Minor changes have been made to the text, and further details have been added to the supplemental information.

Pages 18-19 Lines 327-339 the text now reads: "In addition to the errors discussed above, accuracy in CCN measurements depend on the accuracy of the instrument calibration. Specifically, accurate determination of the percent supersaturation set points within the CCN instrument are dependent on accurate sizing of aerosols entering the CCN, and therefore are dependent on the DMA sizing during CCN calibration. CCN calibrations during two standard compounds, ammonium sulfate and sodium chloride, as described in detail in Rose (2008). Fortunately, if the calibration procedure described by Rose is followed and an optimal DMA aerosol to sheath ratios employed, the uncertainties will be minimal. Specifically, this analysis shows that an aerosol/sheath ratio of 1:10 or 1:20 (Case 1 or 2, respectively) is recommended for all CCN calibrations. This will result in $\kappa_{app}$ uncertainties of less than 1%

for all dry sizes (25 to 200 nm).  However, if CCN calibrations are performed using a DMA operated with less than ideal aerosol to sheath ratios, substantial errors will be introduced. Analysis of the impact of DMA uncertainties on CCN calibrations are discussed in detail in the Supplemental Materials. In the worst case scenario amongst the cases evaluated here (Case 4), the resulting uncertainty in $\kappa_{app}$ is 15%."

Pages 36 Lines 700-703 the text now reads: "By extension, the issue of uncertain sizing by the DMA leads to added uncertainties in the CCN instrument calibrations which are strongly dependent on the chosen aerosol to sheath ration within the DMA.  We recommend conducting all CCN calibrations with DMA aerosol to sheath ratio of 1:10 or 1:20 which will reduce kappa uncertainties to less than 1% for all dry sizes (25 to 200 nm)."

In addition, Figures S3-S5 and the associated text have now been added to the Supplemental Materials.

[Figure]

**Figure S3** CCN data used to determine new parameters during calibration.

[Figure]

**Figure S4** Exemplary $(NH_4)_2SO_4$ CCN activation curves for DMA Cases 1-7, resulting from inaccurate CCN instrument calibration. The true activation curve is shown with dashed lines, and the observed activation (inaccurate supersaturation reported by the instrument) is shown with solid lines.

[Figure]

**Figure S5** (a) Critical supersaturation of ammonium sulfate and sodium chloride particles calculated for DMA Cases 1-7 for sodium chloride (triangles) and ammonium sulfate (circles), following calibration with the same DMA settings. Ammonium sulfate and sodium chloride curves from κ-Köhler theory are shown for comparison. (b) Apparent hygroscopicity $\kappa_{app}$ for DMA Cases 1-7. (c) DMA-flow-derived artifacts in ammonium sulfate $\kappa_{app}$ are shown for each DMA case. (d) DMA-flow-derived artifacts in sodium chloride $\kappa_{app}$ are shown for each DMA case.

Technical Comments

1. Reviewer comment: In figure 3a, in the lognormal distribution it is dN/dlogDp that follows the shape of the curve rather than N (number concentration, y-axis) or dN/dDp.

   Authors' response: Our distribution represents the number concentration.

Authors' response to comments on "Instrument Artifacts Lead to Uncertainties in Parameterizations of Cloud Condensation Nucleation", Revised Submission, Referee #3

Authors' response: We thank the Reviewer for her/his feedback, which we feel have improved the manuscript.
Additional specific modifications are discussed below.

1) Reviewer comment: Line 98-Differences in reported kapp values… 'ed' added

   Authors' response: We thank the reviewer for finding this error.

   Authors' changes to the manuscript: Page 6 Line 95 the text now reads: "Differences in reported $\kappa_{app}$ values…"

2) Reviewer comment: Line 96-Differences in an aerosol's ability… 'an' added

   Authors' response: We thank the reviewer for finding this error.

   Authors' changes to the manuscript: Page 6 Line 99 the text now reads: "…rather any actual differences in an aerosol's ability…"

3) Reviewer comment: Line 643-Operating conditions: lowest DMA/sheath… 'colon and space issue'

   Authors' response: We thank the reviewer for finding this error.

   Authors' changes to the manuscript: Page 35 Lines 664-668 the text now reads: "The lowest combined artifacts ($0.0021 < \kappa_{app,artifact} < 0.0074$, NaCl) occurred as a result of ideal operating conditions: lowest DMA/sheath ratio, corrected multiple particle charging, and little to no undercounting."

4) Reviewer comment: The references are not spaced.

   Authors' response: We thank the reviewer for finding this error.

   Authors' changes to the manuscript: Spaces have been inserted between references.

5) Reviewer comment: Finally, the DMT CCN-100 can be operated under various flows from 200-900 cc/min although it is typically operated at 500 cc/min.  The flow ratio (aerosol to sheath) of the CCNC may also be altered but is typically set at 1:10. The paper should specify these details of operation.

Authors' response: We have added the recommended details to **Section 5. Artifacts derived from cloud condensation nuclei instruments.**

[revised manuscript text omitted]